# Arsenic in drinking water: An analysis of global drinking water regulations and recommendations for updates to protect public health

Seth H. Frisbie[1]*, Erika J. Mitchell[2]

1 Department of Chemistry and Biochemistry, Norwich University, Northfield, Vermont, United States of America, 2 Better Life Laboratories, Inc., East Calais, Vermont, United States of America

* sfrisbie@norwich.edu

**Data Availability Statement:** All relevant data are within the paper and its Supporting Information files.

**Funding:** SHF received support for this project from the Norwich University Board of Fellows

## Abstract

Evidence-based public health policy often comes years or decades after the underlying scientific breakthrough. The World Health Organization's (WHO's) provisional 10 µg/L arsenic (As) drinking water guideline was set in 1993 based on "analytical achievability." In 2011, an additional proviso of "treatment performance" was added; a health-based risk assessment would lead to a lower and more protective guideline. Since the WHO does not require United Nations member states to submit copies of national drinking water regulations, there is no complete database of national drinking water standards or guidelines. In this study, we collated and analyzed all drinking water regulations for As from national governments worldwide. We found regulations for 176 countries. Of these countries, 136 have drinking water regulations that specify 10 µg/L As or less, while 40 have regulations that allow more than 10 µg/L of As; we could not find any evidence of regulations for 19 countries. The number of people living in countries that do not meet the WHO's guideline constitutes 32% of the global population. Global As regulations are also strongly tied to national income, with high income countries more likely to meet the WHO's guideline. In this study, we examined the health risk assessments that show a clear need for reducing As exposure to levels far below the current WHO provisional guideline. We also show that advances in analytical chemistry, drinking water treatment, and the possibility of accessing alternative drinking water supplies without As suggest that both low-income countries with limited resources and high-income countries with adequate resources can adopt a lower and more protective national drinking water standards or guidelines for As. Thus, we recommend that regulators and stake holders of all nations reassess the possibilities for improving public health and reducing health care expenses by adopting more stringent regulations for As in drinking water.

## 1. Introduction

Arsenic (As) is a common drinking water contaminant that is often found in groundwater wells [1–6]. Even at very low concentrations, chronic consumption of As in drinking water has

Faculty Development Prize. There is no grant number for this support. The Board of Fellows had no role in study design, data collection and analysis, decision to publish, or preparation of the manuscript.

**Competing interests:** The authors have declared that no competing interests exist. EJM's affiliation is with Better Life Laboratories, a nonprofit organization that conducts scientific research and provides technical expertise, equipment, and training to help needy people around the world. Better Life Laboratories received no specific funding for this project from any donors. Donors to Better Life Laboratories provided no input in choosing the subject matter of this project, the hypotheses that were tested, the method of analysis, the research findings, or the manner of disseminating the results.

been strongly associated with a variety of cancers and other adverse health effects in humans [7–13]. At least 226 million people in 56 countries are exposed to unsafe concentrations of As in drinking water and food [14].

It often takes years or decades for an advance in science to cause a change in public health policy [15]. Notably, we will demonstrate that all national standards and guidelines for As in drinking water are based on outdated assumptions. For example, many low-and middle-income countries still use the World Health Organization's (WHO's) 1963 drinking water standard for As of 50 micrograms per liter (µg/L), even though the WHO lowered its recommended maximum concentration for As to 10 µg/L in 1993 [16, 17]. Most high-income countries use a 10 µg/L As drinking water standard, consistent with the current WHO recommended maximum concentration. However, many regulators and stakeholders may not be aware that the WHO 10 µg/L guideline is deemed "provisional" [17, 18]. Although risk assessment data indicate a lower guideline would be more appropriate, the WHO retains the 10 µg/L for As based on an assumed practical quantification limit of routine laboratories, as well as an assumed practical drinking water treatment limit [17, 18]. Much new technology for the testing and treatment of As has been developed since the provisional 10 µg/L WHO guideline was first set based on practical concerns rather than health data, but the provisional guideline value has not been updated since 1993 [17, 18].

This causes a significant threat to global public health. Current research shows that more protective national standards and guidelines for As in drinking water are technologically feasible and urgently necessary to protect public health. In this study we collate all current national standards for As in drinking water into a database that can be used to examine patterns of As regulations based on geographic regions and income. We review the basis of the WHO's drinking water guideline for As and why it is deemed "provisional". We examine risk assessment data that indicate that an As guideline of 10 µg/L does not provide sufficient protection against cancer and other adverse health effects. Finally, we review current quantification and treatment technologies, demonstrating the technological feasibility of reducing the maximum allowable concentration of As in drinking water in both high and low income countries.

## 2. Materials and methods

### 2.1 International drinking water standards for arsenic

In order to better understand the state of current regulations for As in drinking water worldwide, we collated all available national standards. We began with a list of the 193 United Nations member states and supplemented this list with 2 other states for which the World Bank provides income group data, Taiwan and Kosovo [19, 20]. We then searched for official laws or decrees promulgated by these national governments regulating As in drinking water. For each country we began our search for national laws and decrees on As in drinking water at the Law Library of Congress website (http://www.loc.gov/law/help/guide/nations.php). When possible, the official online government gazette of a country was also used for this search. If necessary, the FAOLEX database of national laws and regulations on food, agriculture, and renewable natural resources [21], Google Scholar [22], and Google were also used for this search. We also searched catalogs of national standards agencies. When necessary, Google Translate [23] was used to translate between English and the official language of a country. Common search terms included the name of the country, the official gazette of the country, "drinking water quality standards", "drinking water standards", "drinking water", "water", "arsenic", "µg/L", and "mg/L" in both English and the official language of the country. We also compared our search results with those of previous partial surveys of international drinking water regulations [24–27].

For countries in which our search methods could not locate a national law or decree, we continued our search, seeking secondary evidence for regulations such as peer-reviewed articles, dissertations, theses, or similar documents that state a drinking water standard for As in a country. Secondary evidence of regulations was only used when primary evidence was not found. We searched for secondary evidence of regulations using the name of the country, the official gazette of the country, "drinking water quality standards", "drinking water standards", "drinking water", "water", "arsenic", "μg/L", and "mg/L" in both English and the official language of the country. We also used internet searches to determine the national agency or organization responsible for setting drinking water quality standards in these countries and contacted representatives of these agencies via email and Facebook in a national language requesting help obtaining the standards.

If we were unable to find either primary or secondary evidence for a drinking water regulation for As in a country, we assumed that there was most likely no national standard or guideline. There were 19 countries for which we could not find any evidence of drinking water regulations for As.

To understand the relationships between population, income, and As standards, we collated population, gross domestic product (GDP), and GDP per capita data for each country [20, 28, 29]. We selected 2019 population and GDP data since it is the most current data before the demographic and economic upheavals caused by the COVID-19 pandemic. Although COVID-19 was first detected during 2019, as of January, 2020, international economists were not yet expecting it to have a major impact on the world economy [30].

## 2.2 Data analysis and statistics

We used R version 4.1.1, "Kick Things", released August 10, 2021 to calculate descriptive statistics and perform hypothesis testing. For hypothesis testing, we assumed a 95% confidence level for significance. Multiple statistical comparisons of the data were not made, so no corrections for multiple comparisons were applied. Figures and maps were also created with R using the R packages ggplot2 and ggmaps.

## 3. Results and discussion

The results of our search for national As regulations are listed in Table 1. In addition to the name of each country and concentration of As specified in the regulation, we have also included the 2019 Gross Domestic Product (GDP) per capita from the World Bank [20], the 2019 WB income category [31], whether the stated standard is determined independently by the country or tied to international regulations or guidelines such as the WHO drinking water guideline, and when the regulation was most recently updated.

## 3.1 Arsenic regulations

We found evidence of regulations for a maximum allowable concentration of As in drinking water for 176 countries. There were 19 countries for which we could not find any evidence of drinking water regulations. By comparison, in a 2015 survey of global drinking water regulations, the WHO found regulations for As in 102 out of 104 countries [24] while the International Water Resources Association's 2018 comparison of water quality guidelines included drinking water standards for 10 countries [25].

Of the As regulations that we found, the lowest maximum allowable concentration of As was 1 μg/L, the highest was 500 μg/L, and the mode was 10 μg/L. The lowest maximum allowable concentration for As of 1 μg/L may be a typographical error in the government document establishing this allowable concentration since the other contaminants listed in the document

**Table 1. International drinking water standards for arsenic and the 2019 gross domestic product (GDP) per capita in United States dollars for 195 countries.**

| Country | As standard (µg/L) | Year of publication | GDP / Capita (2019) [20] | WB Income Class (2019) [31] | Regulatory Link | Regulation type[a] |
|---|---|---|---|---|---|---|
| **Africa** | | | | | | |
| Algeria [32] | 10 | 2011 | $3,974 | LM | | Law/Decree |
| Angola [33] | 50 | 2011 | $2,791 | LM | | Law/Decree |
| Benin [34] | 50 | 2001 | $1,219 | LM | | Law/Decree |
| Botswana [35–37]b | 10 | 2009 | $7,961 | UM | | Standards Org. |
| Burkina Faso [38] | 10 | 2005 | $787 | Low | WHO: 1996 | Law/Decree |
| Burundi [39, 40] | 10 | 2000 | $261 | Low | EAS: 2000 | Standards Org. |
| Cameroon [18, 41] | 10 | 2007 | $1,507 | LM | WHO | Gov. Org. |
| Cape Verde [42] | 10 | 2004 | $3,604 | LM | | Law/Decree |
| Central African Republic [18, 43] | 10 | 2017 | $468 | Low | WHO | Law/Decree |
| Chad [44] | 10 | 2010 | $710 | Low | | Law/Decree |
| Comoros [45] | 50 | 1994 | $1,370 | LM | WHO | Law/Decree |
| Congo | NA | | $2,280 | LM | | |
| Democratic Republic of the Congo | NA | | $581 | Low | | |
| Djibouti [46] | 50 | 2001 | $3,415 | LM | | Law/Decree |
| Egypt [47, 48] | 10 | 2007 | $3,019 | LM | | Gov. Org. |
| Equatorial Guinea | NA | | $8,132 | UM | | |
| Eritrea | NA | | NA | Low | | |
| Ethiopia [49] | 10 | 2013 | $856 | Low | | Standards Org. |
| Gabon [50] | 50 | 2011 | $7,767 | UM | | Gov. Org. |
| Ghana [51–53] | 10 | 2017 | $2,202 | LM | | Standards Org. |
| Guinea [54] | 10 | 1997 | $963 | Low | | Law/Decree |
| Guinea-Bissau | NA | | $697 | Low | | |
| Ivory Coast [18, 55] | 10 | 2017 | $2,276 | LM | WHO | Law/Decree |
| Kenya [56–58] | 10 | 2018 | $1,817 | LM | | Standards Org. |
| Lesotho [59] | NA | | $1,118 | LM | | |
| Liberia [18, 60] | 10 | 2017 | $622 | Low | WHO | Gov. Org. |
| Libya [61–63] | 10 | 2015 | $7,686 | UM | | Standards Org. |
| Madagascar [64] | 50 | 2004 | $523 | Low | | Law/Decree |
| Malawi [65–67] | 50 | 2013 | $412 | Low | | Standards Org. |
| Mali [68] | 10 | 2007 | $879 | Low | | Gov. Org. |
| Mauritania [69] | 10 | 2015 | $1,679 | LM | WHO | Gov. Org. |
| Mauritius [70] | 10 | 1996 | $11,099 | High | | Law/Decree |
| Morocco [71–73] | 10 | 2006 | $3,282 | LM | | Standards Org. |
| Mozambique [74] | 10 | 2004 | $504 | Low | | Law/Decree |
| Namibia [75, 76] | 300 | 1988 | $4,957 | UM | | Law/Decree |
| Niger [18,77] | 10 | 2017 | $554 | Low | WHO | Law/Decree |
| Nigeria [78] | 10 | 2015 | $2,230 | LM | | Standards Org. |
| Rwanda [79, 80] | 10 | 2014 | $820 | Low | | Standards Org. |
| São Tomé and Príncipe [81] | NA | | $1,947 | LM | | |
| Senegal [18, 82] | 10 | 1996 | $1,447 | LM | | Gov. Org. |
| Seychelles [83] | 10 | 2012 | $17,448 | High | | Law/Decree |
| Sierra Leone [84] | NA | | $528 | Low | | |
| Somalia | NA | | NA | Low | | |
| South Africa [85, 86] | 10 | 2015 | $6,001 | UM | | Law/Decree |
| South Sudan [87] | 50 | 2011 | | Low | | Gov. Org. |

*(Continued)*

**Table 1.** (Continued)

| Country | As standard (µg/L) | Year of publication | GDP / Capita (2019) [20] | WB Income Class (2019) [31] | Regulatory Link | Regulation type[a] |
|---|---|---|---|---|---|---|
| Sudan [88, 89] | 7 | 2009 | $713 | Low | | Standards Org. |
| Swaziland [85, 86, 90, 91] | 10 | 2015 | $4,090 | LM | | Standards Org. |
| Tanzania [92–94] | 10 | 2018 | $1,089 | LM | EAS | Standards Org. |
| The Gambia [95] | 50 | 2008 | $778 | Low | | Gov. Org. |
| Togo [96] | 10 | 2015 | $679 | Low | | Gov. Org. |
| Tunisia [97] | 10 | 2013 | $3,317 | LM | | Standards Org. |
| Uganda [98] | 10 | 2014 | $794 | Low | EAS | Standards Org. |
| Zambia [99] | 10 | 2010 | $1,305 | LM | | Standards Org. |
| Zimbabwe [100–104] | 10 | 2014 | $1,464 | LM | | Standards Org. |
| **The Americas** | | | | | | |
| Antigua and Barbuda [105]c | 10 | 2003 | $17,113 | High | CARICOM | Law/Decree |
| Argentina [107] | 10 | 2019 | $9,912 | UM | | Law/Decree |
| Bahamas [108]c | 50 | 2010 | $34,864 | High | CARICOM | Standards Org. |
| Barbados [109]c | 10 | 2017 | $18,148 | High | CARICOM | Gov. Org. |
| Belize [110]c | NA | | $4,815 | UM | CARICOM | |
| Bolivia [111] | 10 | 2018 | $3,552 | LM | | Gov. Org. |
| Brazil [112] | 10 | 2021 | $8,655 | UM | | Law/Decree |
| Canada [113] | 10 | 2020 | $46,195 | High | | Gov. Org. |
| Chile [114] | 10 | 2007 | $14,896 | High | | Law/Decree |
| Colombia [115] | 10 | 2007 | $6,429 | UM | | Gov. Org. |
| Costa Rica [116] | 10 | 2015 | $12,244 | UM | | Law/Decree |
| Cuba [117] | 50 | 2017 | NA | UM | | Standards Org. |
| Dominica [18, 118]c | 10 | 2017 | $8,111 | UM | CARICOM | Gov. Org. |
| Dominican Republic [119] | 50 | 2001 | $8,282 | UM | | Gov. Org. |
| Ecuador [120] | 100 | 2015 | $6,184 | UM | | Law/Decree |
| El Salvador [121] | 10 | 2009 | $4,187 | LM | | Standards Org. |
| Grenada [122]c,d | 1 | 2005 | $10,809 | UM | CARICOM | Law/Decree |
| Guatemala [124] | 10 | 2013 | $4,620 | UM | | Standards Org. |
| Guyanac | NA | | $6,610 | UM | CARICOM | |
| Haiti [125]c | 10 | 2017 | $1,272 | Low | WHO | Gov. Org. |
| Honduras [126] | 10 | 2007 | $2,575 | LM | | Law/Decree |
| Jamaica [127]c | NA | | $5,582 | UM | CARICOM | |
| Mexico [128]e | 10 | 2019 | $10,069 | UM | | Law/Decree |
| Nicaragua [129] | 50 | 2000 | $1,913 | LM | | Law/Decree |
| Panama [130]f | 10 | 2007 | $15,731 | High | | Law/Decree |
| Paraguay [131]g | 500 | 2000 | $5,415 | UM | | Law/Decree |
| Peru [133] | 10 | 2017 | $6,978 | UM | | Law/Decree |
| Saint Kitts and Nevisc | NA | | $19,939 | High | CARICOM | |
| Saint Luciac | NA | | $11,611 | UM | CARICOM | |
| Saint Vincent and the Grenadinesc | NA | | $7,458 | UM | CARICOM | |
| Suriname [134]c | NA | | $6,360 | UM | CARICOM | |
| Trinidad and Tobagoc | NA | | $17,398 | High | CARICOM | |
| United States [135] | 10 | 2018 | $65,298 | High | | Gov. Org. |
| Uruguay [136] | 20 | 2010 | $16,190 | High | | Standards Org. |
| Venezuela [137] | 10 | 1998 | NA | UM | | Law/Decree |
| **Asia** | | | | | | |
| Afghanistan [138] | 50 | 2013 | $507 | Low | | Standards Org. |

(*Continued*)

**Table 1.** (Continued)

| Country | As standard (μg/L) | Year of publication | GDP / Capita (2019) [20] | WB Income Class (2019) [31] | Regulatory Link | Regulation type[a] |
|---|---|---|---|---|---|---|
| Armenia [139–141] | 50 | 2005 | $4,623 | UM | | Law/Decree |
| Azerbaijan [142, 143] | 50 | 1985 | $4,794 | UM | CIS | Standards Org. |
| Bahrain [144, 145] | 10 | 2012 | $23,504 | High | GCC | Standards Org. |
| Bangladesh [146] | 50 | 2019 | $1,856 | LM | | Gov. Org. |
| Bhutan [147] | 10 | 2018 | $3,316 | LM | | Gov. Org. |
| Brunei [17, 148] | 10 | 1993 | $31,087 | High | WHO: 1993 | Gov. Org. |
| Cambodia [149] | 50 | 2004 | $1,643 | LM | | Gov. Org. |
| China [150] | 50 | 2006 | $10,217 | UM | | Standards Org. |
| Georgia [151] | 10 | 2014 | $4,698 | UM | | Law/Decree |
| India [152]h | 10 | 2012 | $2,100 | LM | | Standards Org. |
| Indonesia [153] | 10 | 2010 | $4,142 | UM | | Gov. Org. |
| Iran [154] | 10 | 2010 | NA | UM | | Standards Org. |
| Iraq [155, 156] | 10 | 2009 | $5,955 | UM | | Standards Org. |
| Israel [157] | 10 | 2016 | $43,592 | High | | Law/Decree |
| Japan [158] | 10 | 2015 | $40,247 | High | | Gov. Org. |
| Jordan [159] | 10 | 2015 | $4,405 | UM | | Standards Org. |
| Kazakhstan [160] | 50 | 2015 | $9,812 | UM | | Law/Decree |
| Kuwait [145, 161] | 10 | 2011 | $32,000 | High | GCC | Gov. Org. |
| Kyrgyzstan [162] | 50 | 2004 | $1,309 | LM | | Law/Decree |
| Laos [163] | 50 | 2009 | $2,535 | LM | | Gov. Org. |
| Lebanon [164] | 50 | 1999 | $7,584 | UM | | Standards Org. |
| Malaysia [165] | 10 | 2004 | $11,414 | UM | | Gov. Org. |
| Maldives [166] | 10 | 2017 | $10,627 | UM | | Gov. Org. |
| Mongolia [167–169] | 10 | 2018 | $4,340 | LM | | Standards Org. |
| Myanmar [170–172] | 50 | 2014 | $1,408 | LM | | Gov. Org. |
| Nepal [173] | 50 | 2005 | $1,071 | LM | | Gov. Org. |
| North Korea | NA | | NA | Low | | |
| Oman [145,174] | 10 | 2012 | $15,343 | High | GCC | Standards Org. |
| Pakistan [175] | 50 | 2010 | $1,285 | LM | | Gov. Org. |
| Philippines [176] | 10 | 2016 | $3,485 | LM | | Gov. Org. |
| Qatar [145, 177] | 10 | 2014 | $62,088 | High | GCC | Gov. Org. |
| Saudi Arabia [145, 178] | 10 | 2015 | $23,140 | High | GCC | Gov. Org. |
| Singapore [179] | 10 | 2019 | $65,233 | High | | Law/Decree |
| South Korea [180, 181] | 10 | 2015 | $31,846 | High | | Gov. Org. |
| Sri Lanka [182] | 50 | 2019 | $3,853 | LM | | Law/Decree |
| Syria [183–185] | 50 | 2007 | NA | Low | | Standards Org. |
| Taiwan [186] | 10 | 2017 | NA | High | | Gov. Org. |
| Tajikistan [143, 187] | 50 | 1985 | $871 | LM | CIS | Standards Org. |
| Thailand [188, 189] | 50 | 2008 | $7,807 | UM | | Gov. Org. |
| Timor-Leste [18, 190]i | 10 | 2017 | $577 | LM | WHO | Gov. Org. |
| Turkey [191, 192] | 10 | 2019 | $9,127 | UM | | Law/Decree |
| Turkmenistan [193–195] | NA | | NA | UM | | |
| United Arab Emirates [145, 196]j | 10 | 2014 | $43,103 | High | GCC | Gov. Org. |
| Uzbekistan [197] | 50 | 2006 | $1,725 | LM | | Law/Decree |
| Vietnam [198] | 10 | 2009 | $2,715 | LM | | Gov. Org. |
| Yemen [199] | 10 | 1999 | $774 | Low | | Gov. Org. |
| **Australia and Oceania** | | | | | | |

(*Continued*)

**Table 1.** (Continued)

| Country | As standard (µg/L) | Year of publication [20] | GDP / Capita (2019) [20] | WB Income Class (2019) [31] | Regulatory Link | Regulation type[a] |
|---|---|---|---|---|---|---|
| Australia [200] | 10 | 2017 | $55,060 | High | | Gov. Org. |
| Federated States of Micronesia [135, 201] | 10 | 2018 | NA | LM | US EPA | Law/Decree |
| Fiji [202, 203]k | 10 | 2011 | $6,176 | UM | | Standards Org. |
| Kiribati [18, 24] | 10 | 2017 | $1,655 | LM | WHO | Gov. Org. |
| Marshall Islands [205] | 50 | 1994 | NA | UM | | Law/Decree |
| Nauru [18, 24]l | 10 | 2017 | $11,724 | High | WHO | Gov. Org. |
| New Zealand [207] | 10 | 2018 | $42,745 | High | | Gov. Org. |
| Palau [208] | 50 | 1996 | $14,902 | High | | Gov. Org. |
| Papua New Guinea [209] | 50 | 2006 | $2,829 | LM | | Law/Decree |
| Samoa [210–212] | 10 | 2016 | $4,209 | UM | | Gov. Org. |
| Solomon Islands [18, 213] | 10 | 2017 | $2,374 | LM | WHO | Law/Decree |
| Tonga [18, 24, 214]m | 10 | 2017 | $4,903 | UM | WHO | Gov. Org. |
| Tuvalu [18, 215] | 10 | 2017 | $4,059 | UM | WHO | Gov. Org. |
| Vanuatu [216] | 10 | 2019 | $3,115 | LM | | Law/Decree |
| **Europe** | | | | | | |
| Albania [217] | 50 | 2016 | $5,353 | UM | | Law/Decree |
| Andorra [218]n | 10 | 2007 | $40,886 | High | | Law/Decree |
| Austria [219] | 10 | 2001 | $50,138 | High | EU | Law/Decree |
| Belarus [220] | 10 | 2015 | $6,663 | UM | | Law/Decree |
| Belgium [221] | 10 | 2003 | $46,421 | High | EU | Law/Decree |
| Bosnia and Herzegovina [222] | 10 | 2010 | $6,109 | UM | | Law/Decree |
| Bulgaria [223] | 10 | 2001 | $9,828 | UM | EU | Law/Decree |
| Croatia [224] | 10 | 2019 | $14,936 | UM | EU | Law/Decree |
| Cyprus [225] | 10 | 2001 | $20,815 | High | EU | Law/Decree |
| Czech Republic [226] | 10 | 2014 | $23,495 | High | EU | Law/Decree |
| Denmark [227] | 5 | 2015 | $60,170 | High | EU | Law/Decree |
| Estonia [228] | 10 | 2002 | $23,723 | High | EU | Gov. Org. |
| Finland [229] | 10 | 2014 | $48,783 | High | EU | Gov. Org. |
| France [230, 231] | 10 | 2017 | $40,494 | High | EU | Law/Decree |
| Germany [232] | 10 | 2017 | $46,445 | High | EU | Law/Decree |
| Greece [233] | 10 | 2017 | $19,583 | High | EU | Law/Decree |
| Hungary [234] | 10 | 2002 | $16,732 | High | EU | Law/Decree |
| Iceland [235] | 10 | 2001 | $66,945 | High | | Law/Decree |
| Ireland [236] | 10 | 2014 | $78,661 | High | EU | Law/Decree |
| Italy [237, 238] | 10 | 2016 | $33,228 | High | EU | Gov. Org. |
| Kosovo [239] | 10 | 2012 | $4,345 | UM | EU | Law/Decree |
| Latvia [240] | 10 | 2017 | $17,829 | High | EU | Law/Decree |
| Liechtenstein [241] | 10 | 2018 | NA | High | | Law/Decree |
| Lithuania [242] | 10 | 2017 | $19,602 | High | EU | Law/Decree |
| Luxembourg [243] | 10 | 2017 | $114,705 | High | EU | Law/Decree |
| Malta [244] | 10 | 2009 | $29,821 | High | EU | Law/Decree |
| Moldova [245] | 10 | 2019 | $4,504 | LM | | Law/Decree |
| Monaco [246] | 10 | 2017 | | High | | Law/Decree |
| Montenegro [247] | 10 | 2012 | $8,909 | UM | | Law/Decree |
| Netherlands [248] | 10 | 2011 | $52,331 | High | EU | Law/Decree |

(Continued)

**Table 1.** (Continued)

| Country | As standard (µg/L) | Year of publication | GDP / Capita (2019) [20] | WB Income Class (2019) [31] | Regulatory Link | Regulation type[a] |
|---|---|---|---|---|---|---|
| Norway [249] | 10 | 2016 | $75,420 | High | | Law/Decree |
| Poland [250] | 10 | 2017 | $15,693 | High | EU | Law/Decree |
| Portugal [251] | 10 | 2017 | $23,252 | High | EU | Law/Decree |
| Republic of North Macedonia [252] | 10 | 2018 | $5,954 | UM | | Law/Decree |
| Romania [253] | 10 | 2019 | $12,920 | High | EU | Law/Decree |
| Russia [254] | 50 | 2001 | $11,774 | UM | | Gov. Org. |
| San Marino [255] | 10 | 2012 | NA | High | | Law/Decree |
| Serbia [256] | 10 | 2019 | $7,412 | UM | | Law/Decree |
| Slovakia [257] | 10 | 2006 | $19,266 | High | EU | Law/Decree |
| Slovenia [258] | 10 | 2015 | $25,946 | High | EU | Law/Decree |
| Spain [259] | 10 | 2003 | $29,367 | High | EU | Law/Decree |
| Sweden [260] | 10 | 2017 | $51,615 | High | EU | Law/Decree |
| Switzerland [261] | 10 | 2020 | $81,994 | High | | Gov. Org. |
| Ukraine [262] | 10 | 2010 | $3,465 | LM | | Gov. Org. |
| United Kingdom [263]o | 10 | 2016 | $42,330 | High | | Law/Decree |

Abbreviations: CARICOM = Caribbean Community; CIS = Commonwealth of Independent States; EAS = East Africa States; EU = European Union; GCC = Gulf Cooperation Council; Gov. Org. = governmental organization (ministry, agency, department); GDP = gross domestic product; LM = Lower Middle; NA = NA; WHO = World Health Organization; UM = Upper Middle; US EPA (United States Environmental Protection Agency)

[a] Law/Decree includes legislation and presidential or royal decrees; Gov. Org. includes ministries and governmental agencies; Standards Org. includes standards bureaus and standards agencies.

[b] Standard withdrawn October 14, 2016 [36].

[c] Caribbean Community and Common Market (CARICOM) standard for packaged purified drinking water is 50 µg/L [106].

[d] The official government document lists this value as "1,0 ug/l", which is a magnitude lower than the contemporaneous WHO value [122, 123]. Since the other water contaminants follow the WHO values, this value is likely a typographical error for "10 ug/l".

[e] As standard only currently applies to municipalities of > 500K inhabitants; to municipalities of > 50K inhabitants by 2022 and to all by 2025 [128].

[f] Applies specifically to bottled water [130].

[g] The official government document lists this value as "0.5 mg/l", which is a magnitude higher than a previous WHO value [131, 132]. Since the document references the WHO guidelines, and a former WHO guideline for As was 50 µg/L, this value is likely a typographical error for "0.05 mg/l".

[h] Permissible limit in the absence of alternative source: 50 µg/L [152].

[i] Author suggests that East Timor uses WHO guidelines as local standards [190].

[j] Standards are specifically for the Emirate of Abu Dhabi [196].

[k] The maximum contaminant level for As in bottled water is 50 µg/L [204].

[l] A Nauru government document states that there are no standards [206], but the 2018 WHO survey noted that Nauru uses WHO guidelines as standards [18, 24].

[m] A secondary source states that there are no national standards [214], but the 2018 WHO survey noted that Tonga uses WHO guidelines as standards [18, 24].

[n] The decree states that in some cases 50 µgl/L is the maximum allowable threshold taking into account the detection limit of the analytical equipment [218].

[o] Applies specifically to private water supplies [263].

followed the values listed in the WHO's drinking water guidelines at the time [122, 123]. The highest maximum allowable concentration of 500 µg/L was also likely a typographical error in the official government record [131]. If these 2 likely errors are set aside, the lowest maximum concentration for As was 5 µg/L and the highest maximum concentration was 300 µg/L. When the likely typographical errors are adjusted to their probable intended values, 2 countries have a maximum allowable concentrations less than 10 µg/L, 134 countries have maximum allowable concentrations of 10 µg/L, and 40 countries have maximum allowable concentrations of greater than 10 µg/L. For all statistical analyses that follow, we retained the published regulatory values without adjustments for possible typographical errors.

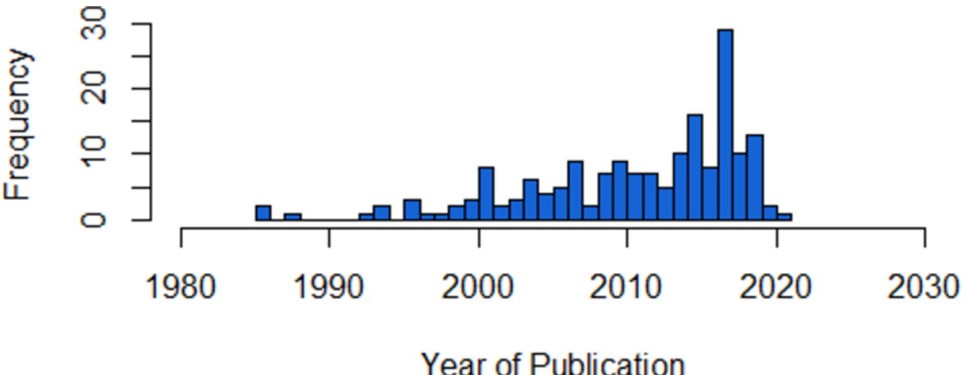

**Fig 1. Comparison of publication dates of arsenic drinking water standards.**

The oldest regulation for As in drinking water currently in force is dated 1985 [142, 143, 187], while the most recent regulation dates to 2021 [112]. The average date of the regulations is 2011, while the mode is 2017 (Fig 1). Arsenic standards are established by national legislation or decree in 85 countries, ministries or agencies in 58 countries, and national standards boards in 33 countries. Many standards published by national standards boards are copyrighted documents and must be purchased or licensed for a fee in order to be accessed.

## 3.2 Arsenic regulations, population, and per capita income

Seventy percent (136) of the 195 countries in our survey have regulations for As in drinking water that are equal to or more protective than the WHO's drinking water guideline of 10 μg/L, while 21% (40) of the 195 countries have regulations that are less protective than the WHO's 10 μg/L guideline, and we could not find regulations for 10% (19) of the 195 countries. Sixty-six percent of the world's population lives in countries with As drinking water regulations equal to or more protective than the WHO's 10 μg/L guideline, 32% live in countries with regulations that are less protective than the WHO's 10 μg/L guideline, and 2% live in countries where we could not find evidence of a drinking water guideline for As (Fig 2).

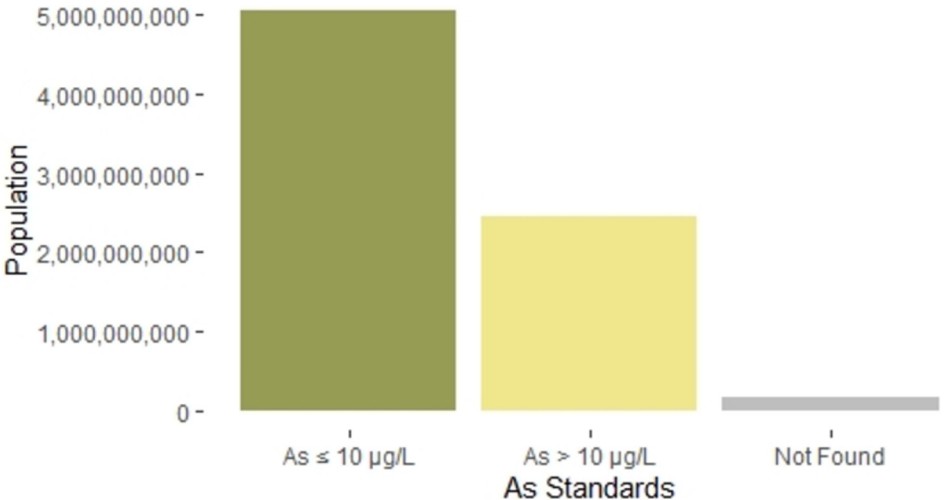

**Fig 2. Comparison of populations covered by different levels of arsenic drinking water standards.**

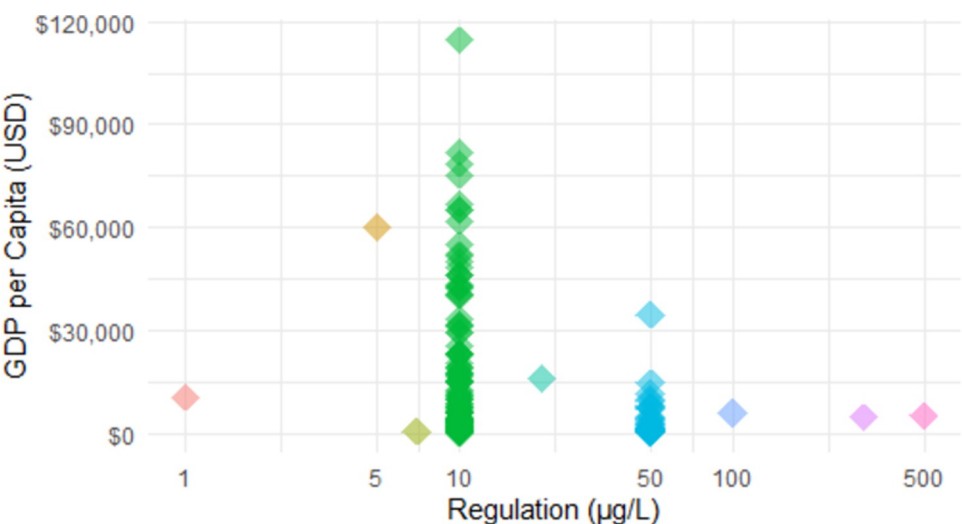

**Fig 3. Gross domestic product (GDP) per capita in current (2019) United States dollars (US $) versus national drinking water standard or guideline for arsenic in micrograms per liter (μg/L) for 180 countries [20].**

Arsenic regulations are also strongly tied to national income as represented by GDP per capita. The sum of all GDPs of the countries with 2019 GDP data and an As regulation equal to or more protective than the WHO guideline of 10 μg/L divided by the sum of the population of these countries was $13,587 per capita. In contrast, the sum of all GDPs of countries with 2019 GDP data and an As regulation less protective than the WHO guideline divided by the sum of the population of these countries was $7,601 per capita. The sum of all GDPs of countries for which we had 2019 GDP data but could not find an As regulation divided by the sum of the population of these countries was $1,227 per capita. The GDPs per capita of countries with As regulations equal to or more protective than the WHO guideline of 10 μg/L were significantly higher ($n = 129$, $M = \$17,678$) than those of countries with As regulations less protective than the current 10 μg/L WHO guideline ($n = 36$, $M = \$5,384$) ($F_{(2,177)} = 7.55$, $p < .001$).

A graph of GDP per capita versus national drinking water standard or guideline for As is shown in Fig 3. Maps of GDP per capita and national drinking water standards or guidelines for As are shown in Figs 4 and 5, respectively.

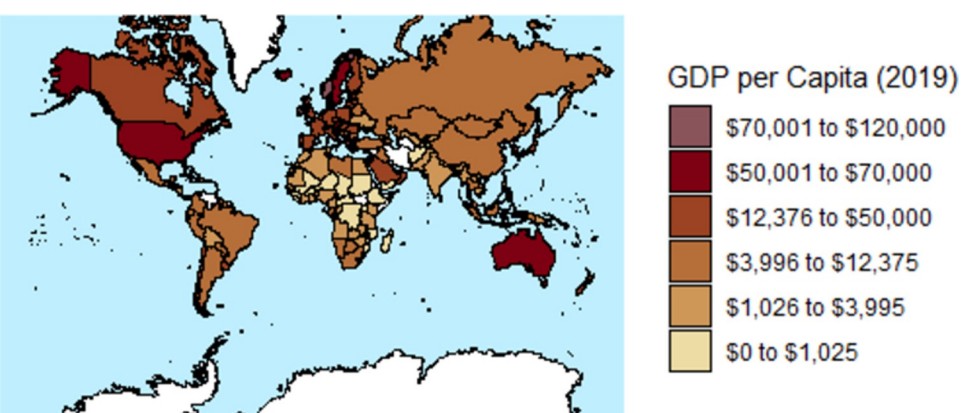

**Fig 4. A map of 2019 gross domestic product (GDP) per capita in United States dollars (US $) for 180 countries (map base: [264]).**

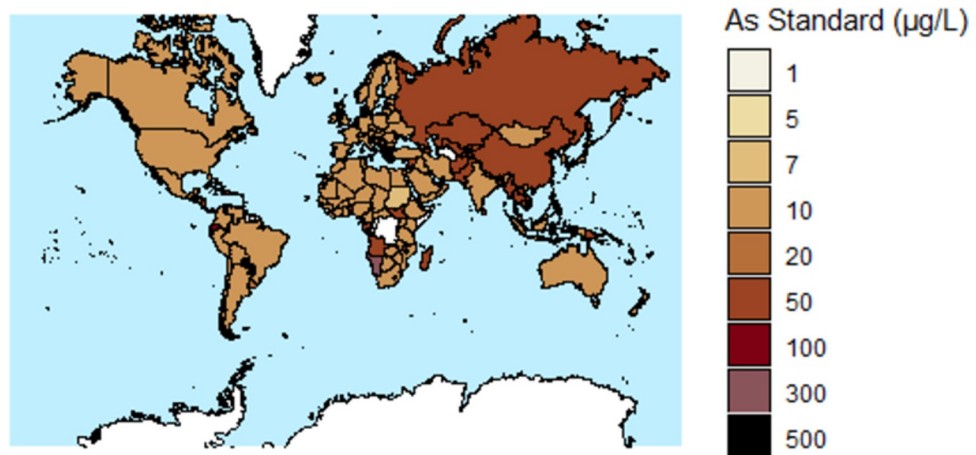

**Fig 5. A map of national drinking water standards or guidelines for arsenic in micrograms per liter (µg/L) for 180 countries (map base: [264]).**

The World Bank classifies economies into one of 4 groups: low income, lower middle income, upper middle income, and high income [31]. We found no difference in recency of As regulations by income group ($F_{(3,175)} = 0.63$, $p = 0.60$). For the countries that have As regulations, the mean As regulation for low income countries was 21 µg/L, for lower middle income countries the mean was 24 µg/L, for upper middle income countries the mean was 38 µg/L, and for high income countries the mean was 11 µg/L (Fig 6). An analysis of variance (ANOVA) on these means yielded significant variation between income classes ($F_{(3, 172)} = 3.15$, $p = .03$). A post hoc Tukey test showed that the mean As standard of the high income class differed significantly from that of the upper middle income class ($p = .01$) while the remaining differences were not significant ($p > 0.05$).

Sixteen countries tie their maximum allowable As concentrations directly to the WHO drinking water guidelines. A large number of other countries have maximum allowable As concentrations that equal the current or previous WHO drinking water guidelines. This

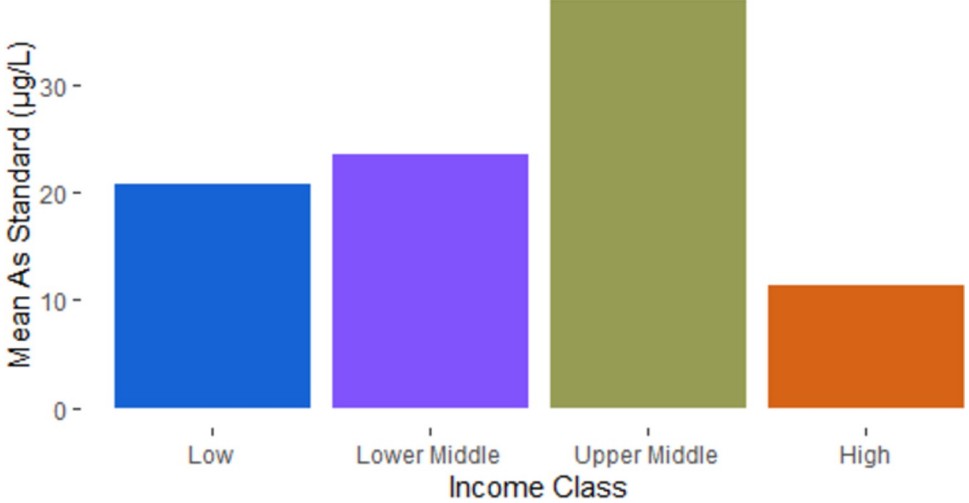

**Fig 6. Comparison of mean national arsenic drinking water standards by income group.**

underscores the importance of the WHO drinking water guidelines for protecting the health of world citizens.

### 3.3 The drinking water guideline set by the World Health Organization

The WHO first established a drinking water standard for As, 200 μg/L, in 1958 based on health concerns [123, 265]. As adverse health effects from As exposure received more study, this As drinking water standard was lowered to 50 μg/L in 1963 [16, 123]. In 1971, the World Health Organization (WHO) deemed its 50 μg/L As standard a "tentative limit" [132] and noted, "Some epidemiological studies have suggested that arsenic is carcinogenic but no real proof of the carcinogenicity to man of arsenic in drinking-water has been forthcoming. It would seem wise to keep the level of arsenic in drinking-water as low as possible" [132].

In 1971, 50 μg/L was "as low as possible" for a limit because the available methods for the routine analysis of As in drinking water were not accurate, precise, sensitive, or affordable enough to reliably measure lower concentrations [132]. For example, the recommended method of polarographic estimation was often not accurate due to interferences, and the incomplete reduction of analyte [132, 266]. Similarly, the commonly used spectrophotometric determination of As by the silver diethyldithiocarbamate ($AgSCSN(CH_2CH_3)_2$) method was not precise due to the sometimes incomplete generation of arsine gas ($AsH_3(g)$) from the sample matrix [132, 267]. In contrast, atomic absorption spectroscopy (AAS) was accurate, precise, and sensitive, but not affordable in many low-income countries [132, 267].

In 1984, the WHO began publishing drinking-water "guidelines" instead of "standards" [268]. Guidelines were intended to provide guidance to national regulators and stakeholders, but they are specifically not to be taken as international standards. National regulators are encouraged to take local conditions, resources, and hazards into account when setting national standards [268, 269]. The 1984 WHO drinking-water guideline for As was maintained at 50 μg/L, the former WHO standard [268].

In 1993, the WHO replaced its earlier "tentative limit" of 50 μg/L with a "provisional guideline" of 10 μg/L for As in drinking water [17]. By this time, a skin cancer risk in humans was known, and other cancer risks were suspected. "Inorganic arsenic is a documented human carcinogen and has been classified by IARC [International Agency for Research on Cancer] in Group 1 ['This category is used when there is *sufficient evidence* of carcinogenicity in humans.']. A relatively high incidence of skin and possibly other cancers that increase with dose and age has been observed in populations ingesting water containing high concentrations of arsenic" [17]. The WHO calculated a health-based value of 0.17 μg/L, but noted that this value was below the practical quantification limit of 10 μg/L [17]. Therefore, the 10 μg/L drinking water guideline for As was initially "provisional" because of what is now called "analytical achievability" [17, 269]. Thus, the 1993 WHO 10 μg/L drinking water guideline for As was based on practical analytical concerns rather than health data, since health data would have led to a lower guideline [17].

The WHO has maintained this 10 μg/L provisional drinking water guideline for As in all subsequent editions and addendums of *Guidelines for Drinking-water Quality* since 1993 [18, 123, 270–272]. In 2006, the WHO stated that As in drinking water not only causes skin cancer, but also causes bladder and lung cancers [271]. "There is overwhelming evidence from epidemiological studies that consumption of elevated levels of arsenic through drinking-water is causally related to the development of cancer at several sites, particularly skin, bladder and lung" [271]. In 2011, the WHO added kidney cancer to this list [272]. "The International Programme on Chemical Safety (IPCS) concluded that long-term exposure to arsenic in drinking-water is causally related to increased risks of cancer in the skin, lungs, bladder and kidney, as

well as other skin changes, such as hyperkeratosis and pigmentation changes" [272]. However, the WHO did not lower their drinking water guideline for As "because [the] calculated guideline value is below the achievable quantification level" [272]. Again, "guideline values are not set at concentrations of substances that cannot reasonably be measured. In such circumstances, provisional guideline values are set at the reasonable analytical limits" [269, 272].

In 2011, the WHO added a second provision to its provisional 10 μg/L drinking water guideline for As [272]. This second provision is based on "treatment performance" [272]. More specifically, "It is technically feasible to achieve arsenic concentrations of 5 μg/l or lower using any of several possible treatment methods. However, this requires careful process optimization and control, and a more reasonable expectation is that 10 μg/l should be achievable by conventional treatment (e.g. coagulation)" [272]. In 2017, the WHO maintained its 10 μg/L provisional drinking water guideline for As, the "analytical achievability" provision for this guideline, and the "treatment performance" provision for this guideline [18, 269].

In summary, the current WHO 10 μg/L provisional drinking water guideline for As is not set according to a health-based risk assessment, since this would require a lower maximum concentration, but rather, it is based on the detection limits for the routine analysis of As in drinking water, which have not been updated since 1993, and treatment technologies for As, which have not been updated since 2011 [18, 272].

## 3.4 Health-based drinking water guidelines

In 2000, the United States Environmental Protection Agency (U.S. EPA) proposed a non-enforceable health-based goal, or Maximum Contaminant Level Goal (MCLG) for As of 0 μg/L [273]. This 0 μg/L MCLG remains in force [135]. In 1999, the U.S. EPA noted that there was a Practical Quantitation Limit (PCL) for As of 3 μg/L [274]. However, instead of using this PQL as the enforceable standard Maximum Contaminant Level (MCL), 5 μg/L was proposed as a national standard based on a cost/benefit analysis [273]. After this proposed MCL of 5 μg/L was set out for public comment, the MCL was raised to 10 μg/L before being adopted as the national standard, over the objections of the U.S. EPA's own Scientific Advisory Board, who argued for a lower MCL [275, 276].

In contrast, the California Environmental Protection Agency (CalEPA) has set a public health goal (PHG) of 0.004 μg/L for As in drinking water [7, 8]. This PHG is based on human health effects; it is not influenced by detection limits or the performance of drinking water treatment systems. Under California law, a PHG must be based on current scientific evidence and give a negligible risk of adverse health effects over a lifetime of exposure [277]. In addition, a PHG must not consider economic, technical, or other societal factors [277]. As a result, the CalEPA 0.004 μg/L PHG is 2,500 times lower than the WHO provisional drinking water guideline and the U.S. EPA drinking water standard of 10 μg/L.

More specifically, the CalEPA PHG is based on the risk of death from lung and bladder cancers after a lifetime of exposure to As in drinking water [8]. These risks were calculated from epidemiological studies in Taiwan, Chile, and Argentina (Eqs 1, 2, 3, 4, 5 and 6) [8]. Since the risk of death from lung and bladder cancers is greater than that from skin cancer, kidney cancer, and noncancer health effects, the risks from these 3 less significant effects were not factored into the CalEPA PHG (Eqs 1 and 2) [8].

$$\frac{y \text{ Excess Cancer Deaths}}{1,000,000 \text{ People}} = \frac{x \text{ μg}}{\text{L}} \times 0.00024_{75} \qquad (\text{Eq 1})$$

The nonsignificant digits, "75", are shown as a subscript (Eqs 1, 2, 3, 4, 5 and 6). These nonsignificant digits are included in all steps of a calculation to prevent rounding error. The

CalEPA rounded their PHG to 1 significant figure as follows (Eq 2) [8].

$$\frac{0.004\ \mu g}{L}\begin{pmatrix} \text{Rounded to} \\ \text{1 Figure} \end{pmatrix} = \frac{\left(\frac{1\ \text{Excess Cancer Death}}{1,000,000\ \text{People}}\right)}{(0.00024_{75})} \tag{Eq 2}$$

This 0.004 μg/L PHG is estimated to result in 1 excess cancer death in 1,000,000 people (Eq 2) [8]. That is, if 1,000,000 people drank water with 0.004 μg of As/L over their lifetimes, it is estimated that 1 of these 1,000,000 people would die from cancer because of this exposure to As. The excess death could be prevented if the dose of the carcinogen is lowered or eliminated [8].

In contrast, the 10 μg/L value used as the WHO provisional drinking water guideline and the U.S. EPA MCL is estimated to result in 2,500 excess cancer deaths in 1,000,000 people (Eq 3), or in 1 excess cancer death in 400 people (Eq 4).

$$\frac{2,500\ \text{Excess Cancer Deaths}}{1,000,000\ \text{People}}\begin{pmatrix} \text{Rounded to} \\ \text{2 Figures} \end{pmatrix} = \frac{10\ \mu g}{L} \times 0.00024_{75} \tag{Eq 3}$$

$$\frac{1\ \text{Excess Cancer Death}}{400\ \text{People}}\begin{pmatrix} \text{Rounded to} \\ \text{2 Figures} \end{pmatrix} = \frac{10\ \mu g}{L} \times 0.00024_{75} \tag{Eq 4}$$

By comparison, the National Research Council estimated that drinking 10 μg/L of As over a lifetime results in 3,700 excess cancer deaths in 1,000,000 males and 3,000 excess cancer deaths in 1,000,000 females (Eq 3) [8].

The 50 μg/L value still used by many countries as national drinking water standards is estimated to result in 12,000 excess cancer deaths in 1,000,000 people (Eq 5), or 1 excess cancer death in 81 people (Eq 6).

$$\frac{12,000\ \text{Excess Cancer Deaths}}{1,000,000\ \text{People}}\begin{pmatrix} \text{Rounded to} \\ \text{2 Figures} \end{pmatrix} = \frac{50\ \mu g}{L} \times 0.00024_{75} \tag{Eq 5}$$

$$\frac{1\ \text{Excess Cancer Death}}{81\ \text{People}}\begin{pmatrix} \text{Rounded to} \\ \text{2 Figures} \end{pmatrix} = \frac{50\ \mu g}{L} \times 0.00024_{75} \tag{Eq 6}$$

## 3.5 Low-cost methods for improving public health by reducing arsenic exposure

**3.5.1 Advances in inexpensive analytical chemistry methods.** One reason for using a 50 μg/L drinking water standard for As instead of the more protective WHO 10 μg/L guideline is the high cost of atomic absorption spectrometers or other sophisticated instruments for measuring total As to 10 μg/L or lower. However, recent developments in analytical methods now make it possible to quantify As to 10 μg/L or lower without expensive equipment.

**3.5.2 Spectrophotometry.** One example of a low-cost method for quantifying As to 10 μg/L or lower without expensive equipment is the arsenomolybdate method, validated in 2005 [267]. By design, the arsenomolybdate method uses the same equipment as the commonly used silver diethyldithiocarbamate (AgSCSN(CH$_2$CH$_3$)$_2$) method for measuring As [267, 278]. In the arsenomolybdate method, As is removed from the sample by reduction to arsine gas (AsH$_3$(g)), collected in an absorber by oxidation to arsenic acid (H$_3$AsO$_4$), colorized

**Table 2. Common drinking water standards, guidelines, and public health goals for total arsenic (As) in micrograms per liter (μg/L), the detection limits for total As in μg/L by spectrophotometry, and the estimated cancer risks at these concentrations (Eqs 1, 2, 3, 4, 5 and 6).** These cancer risks are in bold font and rounded to 2 figures.

| As concentration | Number of Excess Cancer Deaths 1,000,000 People | 1 Excess Cancer Death Number of People |
|---|---|---|
| **50 μg/L** (drinking water standard common in lower income countries) | **12,000 Excess** Cancer Deaths 1,000,000 People | 1 Excess Cancer Death 81 People |
| **10 μg/L** (WHO provisional drinking water guideline; drinking water standard common in higher income countries) | **2,500 Excess** Cancer Deaths 1,000,000 People | 1 Excess Cancer Death 400 People |
| **7.5 μg/L** (detection limit by spectrophotometry using suspended nanoparticles [286]) | **1,900 Excess** Cancer Deaths 1,000,000 People | 1 Excess Cancer Death 540 People |
| **7 μg/L** (detection limit by spectrophotometry using arsenomolybdate [267]) | **1,700 Excess** Cancer Deaths 1,000,000 People | 1 Excess Cancer Death 580 People |
| **4 μg/L** (detection limit by spectrophotometry using an arsenoantimonomolybdenum blue-malachite green complex [283]) | **990 Excess** Cancer Deaths 1,000,000 People | 1 Excess Cancer Death 1,000 People |
| **4 μg/L** (detection limit by spectrophotometry using suspended microparticles [284]) | **990 Excess** Cancer Deaths 1,000,000 People | 1 Excess Cancer Death 1,000 People |
| **0.5 μg/L** (detection limit by spectrophotometry using suspended nanoparticles [285]) | **120 Excess** Cancer Deaths 1,000,000 People | 1 Excess Cancer Death 8,100 People |
| **0.3 μg/L** (detection limit by spectrophotometry using a molybdoarsenate-malachite green complex [282]) | **74 Excess** Cancer Deaths 1,000,000 People | 1 Excess Cancer Death 13,000 People |
| **0.004 μg/L** (Public Health Goal Set by the California Environmental Protection Agency [8]) | **1.0 Excess** Cancer Death 1,000,000 People | 1 Excess Cancer Death 1,000,000 People |

by a sequential reaction to arsenomolybdate, and quantified by spectrophotometry at 835 nm (nanometers) [267]. The method detection limit is 7 μg/L (Table 2) [267]. This detection limit is intended to equal the "concentration of a substance that can be measured and reported with 99% confidence that the analyte concentration is greater than 0" [279, 280]. In summary, the arsenomolybdate method is more accurate, precise, and environmentally safe than the $AgSCSN(CH_2CH_3)_2$ method; and it is more accurate and affordable than the graphite furnace atomic absorption spectroscopy (GFAAS) method [267].

Other advances in inexpensive analytical chemistry methods include the use of cationic dyes, such as malachite green ($C_6H_5C(C_6H_4N(CH_3)_2)_2^+$), that react with oxyanions, such as derivatized As and derivatized phosphorus (P), to form an ionic solid. These solids are either suspended with a surfactant or dissolved with an organic solvent and measured by spectrophotometry [281–283]. More specifically, one method for the determination of total As uses an arsenoantimonomolybdenum blue-malachite green complex [283]. This complex is suspended with Triton™ X-350 and analyzed at 640 nm [283]. The detection limit is 4 μg/L and is defined as the concentration of standard solution that has a 0.01 absorbance (Table 2) [283]. This method is subject to interferences unless the As is removed from the sample by reduction to $AsH_3$ gas before color development [283]. Another method for the determination of total As uses a molybdoarsenate-malachite green complex [282]. This complex is concentrated by

filtration onto a nitrocellulose membrane filter [282]. This complex and filter are dissolved with 2-methoxyethanol (methyl cellosolve; $CH_3OCH_2CH_2OH$), and the filtrate is analyzed at 627 nm [282]. The detection limit is 0.3 μg/L; however, the criteria used to calculate this detection limit was not given (Table 2) [282]. Interference from phosphate is corrected by a selective reduction procedure, interference from ferric iron (Fe(III)) is corrected by a cation exchange procedure, and interference from silicate is corrected by an acidic digestion procedure [282]. Alternatively, all of these interferences are corrected if As is removed from the sample by reduction to $AsH_3$ gas before color development.

More recently, another cationic dye, ethyl violet ($C(C_6H_4N(CH_2CH_3)_2)_3^+$), was reacted with derivatized oxyanions of As to form suspended particles of ionic complexes [284, 285]. More specifically, one method for the determination of total As uses a molybdoarsenate-ethyl violet complex [284]. This complex forms suspended microparticles and an apparently homogenous blue solution [284]. Prior to color development, interference from phosphate is corrected by an anion exchange procedure, and interference from silica is corrected by reaction with sodium fluoride (NaF) [284]. After color development, the excess dye is converted to a colorless carbinol species in strong acid and the molybdoarsenate-ethyl violet complex is analyzed at 612 nm [284]. This decolorization of excess dye significantly reduces the absorbance of reagent blanks and permits a 4 μg/L detection limit (Table 2) [284]. This detection limit is defined as $3\sigma/m$, "where σ is the standard deviation of 5 measurements of the reagent blank, and $m$ is the slope of the calibration graph" [284]. This method was modified and uses ethyl violet, an isopolymolybdate-iodine tetrachloride complex, and molybdoarsenate to form suspended nanoparticles that are analyzed at 550 nm as a determination of total As [285]. Prior to color development, interference from ferric iron is corrected by reaction with ethylenediaminetetraacetic acid (EDTA), interference from phosphate is corrected by an anion exchange procedure, and interference from silica is corrected by reaction with NaF [285]. The detection limit is 0.5 μg/L (Table 2) [285]. This detection limit is defined as $3\sigma/m$ [285].

Another method for the determination of total As uses a reduction and selective extraction of arsenite, As(III), into an ionic liquid functionalized with gold (Au) nanoparticles that are analyzed visually or potentially with a spectrophotometer [286]. The reducing agent is ascorbic acid ($C_6H_8O_6$) [286]. The ionic liquid is prepared by mixing ultrapure water (in this case, resistivity = 18.3 MΩ·cm), tetradecyl (trihexyl) phosphonium chloride (Cyphos® IL-101; $[C_{14}(C_6)_3P]$ Cl), and Triton™ X-114 [286]. This ionic liquid is functionalized by adding chloroauric acid ($HAuCl_4$) and potassium tetrahydroborate ($KBH_4$) [286]. The functionalized ionic liquid is red in the absence of As(III) and blue in the presence of As(III) [286]. The estimated detection limit by naked eye is 7.5 μg/L (Table 2) [286]. This method is highly selective; 1.0 micromolar (μM) concentrations of $K^+$, $Na^+$, $Ca^{2+}$, $Mg^{2+}$, $Al^{3+}$, $Ni^{2+}$, $Fe^{3+}$, $Cr^{3+}$, $Zn^{2+}$, $Mn^{2+}$, $Pb^{2+}$, $Cd^{2+}$, $Hg^{2+}$, $SO_4^{2-}$, $PO_4^{3-}$, $CO_3^{2-}$, $NO_2^-$, and $SiO_3^{2-}$ do not significantly interfere [286]. Higher, but unspecified concentrations, of $Cl^-$, $NO_3^-$, and $SCN^-$ do not significantly interfere [286].

**3.5.3 Summary of advances in inexpensive analytical chemistry methods.** In summary, no country needs to use the less protective 50 μg/L standard or guideline due to the expense of analytical chemistry methods. There are many affordable methods for measuring total As to the more protective WHO 10 μg/L provisional drinking water guideline, or to concentrations as low as 0.3 μg/L (Table 2).

**3.5.4 Advances in inexpensive drinking water treatment technologies.** Another reason for using a 50 μg/L drinking water standard instead of the more protective WHO 10 μg/L guideline is the high expense of drinking water treatment systems. However, advances in inexpensive drinking water treatment technologies have produced technologies that can now remove As to concentrations that are lower than 10 μg/L [287]. Selected examples of these advances follow.

Batch rectors used optimized pH adjustment, oxidation, coagulation, and filtration to economically remove As from drinking water to "about 5 μg/L" during field trials of 10 households and 6 schools in Assam, India [288]. Sodium bicarbonate ($NaHCO_3$) was used to adjust pH, potassium permanganate ($KMnO_4$) was used to oxidize ambient As(III) and Fe(II) to relatively insoluble As(V) and Fe(III), and iron (III) chloride ($FeCl_3$) was used to coagulate the oxidized As and Fe [288]. The treated water was allowed to settle for 1 to 2 hours [288]. The supernatant was filtered through sand-gravel filters [288]. The households each used a 10 L batch rector, 5 schools each used a 25 L batch reactor, and 1 school used a 200 L batch reactor [288]. The concentration of As in the influent ranged from 100 μg/L to 240 μg/L [288]. The concentration of As in the effluent was less than 5 μg/L if the initial concentration of Fe in the influent was less than 1,000 μg/L, and the concentration of As in the effluent was less than 8 μg/L if the initial concentration of Fe in the influent was greater than 2,500 μg/L [288]. The estimated recurring cost per cubic meter ($m^3$) of treated water was approximately US $0.16 per/$m^3$ [288].

An electrocoagulation batch reactor economically removed As from drinking water to concentrations that were always less than 5 μg/L during a 3.5 month field trial in West Bengal, India [289]. More specifically, electrolytic oxidation of a sacrificial iron (Fe) anode produced Fe(III) precipitates. These Fe(III) precipitates reacted with As in the influent and produced As-Fe(III) precipitates. These As-Fe(III) precipitates aggregated or flocculated together. These flocs were mixed with a small amount of alum and removed by gravitational settling [289]. The concentration of As in the influent was 266 ± 42 μg/L and the average concentration of As in the effluent was 2.1 ± 1.0 μg/L [289]. This batch reactor treated 31,000 L (50 batches) of drinking water during the 3.5 month field trial [289]. The cost of treated water was US $0.83/$m^3$ to US $1.04/$m^3$ [289].

Above ground adsorbent filters economically removed As from drinking water to concentrations that were consistently less than 10 μg/L during field trials of 20 households and 3 schools in West Bengal, India [290]. These filters were made of activated carbon, charcoal, fine granular sand, activated laterite, and raw laterite [290]. Laterites are highly weathered soils that are dominated by clay sized particles of iron hydrous oxides and aluminum hydrous oxides [291]. The laterite was activated by sequential treatment with hydrochloric acid (HCl) and sodium hydroxide (NaOH) [290]. The concentration of As in the influent ranged from 50 μg/L to 500 μg/L [290]. The concentration of As in the effluent was always less than 10 μg/L [290]. These filters have a relatively large As removal capacity, 32.5 milligrams (mg)/g, and as a result have a relatively long service life, at least 5 years [290]. The cost of the treated water was less than US $0.35/$m^3$ [290].

Below ground adsorbent filters economically removed As from drinking water to concentrations that were consistently less than 10 μg/L during field trials of 4 households in Hangjinhouqi County, Inner Mongolia, China [292]. These filters were made by mixing 1 mass unit of a locally abundant limestone with 2 mass units of a locally abundant Fe-mineral (hematite and goethite) [292]. This mixture was placed below ground, around the well screen of a conventional tube well [292]. The limestone likely increased the pH of the groundwater [293]. This increase in pH likely enhanced the oxidation of soluble As(III) with dissolved oxygen ($O_2(g)$) to make insoluble As(V) [294]. This As(V) was removed from the groundwater by precipitation with the dissolved calcium ($Ca^{2+}(aq)$) from the limestone and by adsorption to the surface of the Fe-mineral [292]. The concentration of As in the unfiltered groundwater ranged from 318 μg/L to 635 μg/L [292]. The concentration of As in the filtered groundwater was always less than 10 μg/L [292]. "The filtration system was continuously operated for a total volume of 365,000 L, which is sufficient for drinking water supplying a rural household of 5 persons for 5 years at a rate of 40 L per person per day" [292]. The cost of the treated water was less than US $0.10/$m^3$ [292].

**Table 3. Common drinking water standards, guidelines, and public health goals for total arsenic (As) in micrograms per liter (µg/L), the effluent concentrations of total As in µg/L from water treatment systems used in the low-income world, and the estimated cancer risks at these concentrations (Eqs 1, 2, 3, 4, 5 and 6).** These cancer risks are in bold font and rounded to 2 figures.

| As concentration | Number of Excess Cancer Deaths 1,000,000 People | 1 Excess Cancer Death Number of People |
|---|---|---|
| **50 µg/L** (drinking water standard common in lower income countries) | **12,000 Excess Cancer Deaths** 1,000,000 People | 1 Excess Cancer Death 81 People |
| **10 µg/L** (WHO provisional drinking water guideline; drinking water standard common in higher income countries) | **2,500 Excess Cancer Deaths** 1,000,000 People | 1 Excess Cancer Death 400 People |
| **<10 µg/L** (treatment performance using above ground adsorbent filters [290]; below ground adsorbent filters [292]; groundwater extraction, aeration, and reinjection system [295]) | **< 2,500 Excess Cancer Deaths** 1,000,000 People | < 1 Excess Cancer Death 400 People |
| **5 µg/L** (treatment performance using an optimized pH adjustment, oxidation, coagulation, and filtration in a batch reactor [288]) | **1,200 Excess Cancer Deaths** 1,000,000 People | 1 Excess Cancer Death 810 People |
| **<5 µg/L** (treatment performance using an electrocoagulation batch reactor [289]) | **< 1,200 Excess Cancer Deaths** 1,000,000 People | < 1 Excess Cancer Death 810 People |
| **0.004 µg/L** (Public Health Goal Set by the California Environmental Protection Agency [8]) | **1.0 Excess Cancer Death** 1,000,000 People | 1 Excess Cancer Death 1,000,000 People |

A groundwater extraction, aeration, and reinjection system economically removed As from drinking water to concentrations that were consistently less than 10 µg/L during a village scale field trial in West Bengal, India [295]. More specifically, groundwater was extracted from the aquifer with a submersible electric pump [295]. This groundwater was aerated by spraying it into an above ground plastic tank with "ordinary plastic shower heads" and letting it sit in the tank for at least 30 minutes [295]. The final concentration of $O_2$(aq) in this aerated water ranged from 4 mg/L to 6 mg/L [295]. Approximately, 15% to 20% of this aerated water as reinjected into the aquifer by gravity, and the remaining 80% to 85% of this aerated water was used for drinking [295]. The concentration of As in the influent was not clearly specified [295]. The concentration of As in the effluent was always less than 10 µg/L [295]. The cost of the treated water was US $0.50/m$^3$ [295].

In summary, no country needs to use the less protective 50 µg/L standard or guideline due to the expense of treatment technologies. There are many affordable methods for treating water to reduce As concentrations to lower than 10 µg/L (Table 3).

## 3.6 Methods for improving public health by reducing arsenic exposure that require higher expenditures

**3.6.1 Other advances in analytical chemistry methods.** Some countries use a 10 µg/L drinking water standard or guideline because they assume that this continues to be the limit of quantification for routine analytical chemistry laboratories. However, recent advances in analytical chemistry have developed methods that can be used in routine laboratories with detection limits that are at 0.1 µg/L or less [18, 296]. With these advances 10 µg/L As should no longer be considered the practical limit of quantification, so revised As standards or guidelines could be more protective of public health.

**3.6.2 Inductively coupled plasma-mass spectrometry.** Advances in inductively coupled plasma-mass spectrometry (ICP-MS) give detection limits for total As that are at 0.1 µg/L or

less [18, 296]. Moreover, ICP-MS is commonly used in routine drinking water testing laboratories because it can simultaneously measure the concentrations of almost all of the elements on the periodic table. This reduces cost by allowing each sample to be analyzed for multiple elements in just a few seconds.

An ICP-MS ionizes the As or other analytes in an inductively coupled plasma (ICP) and then separates and quantifies these ions in a mass spectrometer (MS) [296]. The drinking water sample is turned into an aerosol and delivered to an argon (Ar) plasma [296]. This plasma is a flame like object at the top of an ICP torch, and has a sufficient concentration of Ar cations ($Ar^+(g)$) and free electrons ($e^-(g)$) to make the gas electrically conductive (Eq 7) [296].

$$Ar_{(g)} \leftrightarrows Ar^+_{(g)} + e^-_{(g)} \tag{Eq 7}$$

This torch is surrounded by an induction coil [296]. This coil transmits power from a radio frequency (RF) generator to the plasma [296]. This energy maintains the ionization of the Ar gas, and the temperature of the plasma from about 5,500 kelvin (K) to 8,000 K [296]. These temperatures are approximately 2 or 3 times hotter than all flame spectroscopic methods [296]. As a result, these higher temperatures help make ICP-MS more sensitive than flame atomic absorption spectrometry (FAAS), flame atomic emission spectrometry (FAES), and flame atomic fluorescence spectrometry (FAFS) [296]. The reaction for the ionization of As in a plasma follows (Eq 8).

$$As_{(g)} \leftrightarrows As^+_{(g)} + e^-_{(g)} \tag{Eq 8}$$

These $Ar^+$(g; Eq 7) ions, $As^+$(g; Eq 8) ions, and any other ions exit the ICP and enter the MS [296]. An MS is a mass filter and a mass detector [296, 297]. The most common type of MS used in atomic mass spectroscopy is the quadrupole mass analyzer [296]. This analyzer uses direct current (DC) and radiofrequency (RF) fields to filter ions [296, 297]. These ions are directed in between 4 parallel rods, the quadrupole [296, 296]. These rods are connected to a source of variable DC voltages; 2 rods are positively charged and 2 rods are negatively charged [296]. In addition, variable RF alternating current (AC) voltages are applied to each pair of rods [296]. The DC and AC voltages on the rods are simultaneously changed; however, the ratio of these voltages is held constant [296]. This makes the ions oscillate between the rods [296]. At any instant, only the ions with the desired mass and charge exit the quadrupole and are detected at an ion transducer; all other ions hit the rods, are converted to 0 charge, and are not detected [296]. In this way, an entire spectrum of atomic masses can be scanned in less than 0.1 seconds [296]. Ultimately, the MS detector measures the ratio of mass to charge (*m/z*) for each cation [296–297]. Since ions with multiple charges are rarely produced, the charge (*z*) is normally assumed to equal 1 and *m/z* is normally the mass of the cation [296–297].

The only naturally occurring isotope for As is $^{75}$As; that is, all of the As in nature has 75 protons and neutrons [298]. Therefore, all of the As in nature has *m/z* = 75. Unfortunately, if a drinking water sample has chlorine (Cl), calcium (Ca), or sulfur (S), these common atoms can react with the plasma to form $^{40}Ar^{35}Cl^+$(g), $^{38}Ar^{37}Cl^+$(g), $^{37}Cl_2{}^1H^+$(g), $^{40}Ca^{35}Cl^+$(g), or $^{40}Ar^{34}S^1H^+$(g) [299, 300]. These are polyatomic interferences; they all have *m/z* = 75 and cannot be distinguished from $^{75}As^+$(g) [299, 300]. No other isotope of As can be used to eliminate this systematic error [298]. Mathematical corrections can be used to estimate the contribution of these polyatomic interferences to the signal at *m/z* = 75; however, the combined uncertainties in these corrections tend to inflate the detection limit [301].

Fortunately, advances in collision/reaction cell (C/RC) technology are being used to eliminate these polyatomic interferences and give lower detection limits [301, 302]. If a polyatomic ion exits an ICP, enters a C/RC, and collides with an inert gas, such as helium (He(g)), the interference can be eliminated by breaking the polyatomic ion apart (collision-induced

**Table 4. Common drinking water standards, guidelines, and public health goals for total arsenic (As) in micrograms per liter (µg/L), the detection limits for total As in µg/L by inductively coupled plasma-mass spectrometry (ICP-MS), inductively coupled plasma-tandem mass spectrometry (ICP-MS/MS), and hydride generation-gas chromatography-photoionization detection (HG-GC-PID), and the estimated cancer risks at these concentrations (Eqs 1, 2, 3, 4, 5 and 6).** These cancer risks are in bold font and rounded to 2 figures.

| As concentration | Number of Excess $\frac{\text{Cancer Deaths}}{1,000,000 \text{ People}}$ | 1 Excess $\frac{\text{Cancer Death}}{\text{Number of People}}$ |
|---|---|---|
| **50 µg/L** (drinking water standard common in lower income countries) | **12,000 Excess** $\frac{\text{Cancer Deaths}}{1,000,000 \text{ People}}$ | 1 Excess $\frac{\text{Cancer Death}}{81 \text{ People}}$ |
| **10 µg/L** (WHO provisional drinking water guideline; drinking water standard common in higher income countries) | **2,500 Excess** $\frac{\text{Cancer Deaths}}{1,000,000 \text{ People}}$ | 1 Excess $\frac{\text{Cancer Death}}{400 \text{ People}}$ |
| **0.029 µg/L** (detection limit by ICP-MS using a Knotted Reactor [304]) | **7.2 Excess** $\frac{\text{Cancer Deaths}}{1,000,000 \text{ People}}$ | 1 Excess $\frac{\text{Cancer Death}}{140,000 \text{ People}}$ |
| **0.025 µg/L** (detection limit by ICP-MS using a Collision/Reaction Cell [302]) | **6.2 Excess** $\frac{\text{Cancer Deaths}}{1,000,000 \text{ People}}$ | 1 Excess $\frac{\text{Cancer Death}}{160,000 \text{ People}}$ |
| **0.004 µg/L** (public health goal set by the California Environmental Protection Agency [8]) | **1.0 Excess** $\frac{\text{Cancer Death}}{1,000,000 \text{ People}}$ | 1 Excess $\frac{\text{Cancer Death}}{1,000,000 \text{ People}}$ |
| **0.0016 µg/L** (detection limit by ICP-MS/MS using a collision/reaction cell with $O_2(g)$ [305]) | **0.40 Excess** $\frac{\text{Cancer Deaths}}{1,000,000 \text{ People}}$ | 1 Excess $\frac{\text{Cancer Death}}{2,500,000 \text{ People}}$ |
| **0.00082 µg/L** (detection limit by HG-GC-PID [307]) | **0.20 Excess** $\frac{\text{Cancer Deaths}}{1,000,000 \text{ People}}$ | 1 Excess $\frac{\text{Cancer Death}}{4,900,000 \text{ People}}$ |
| **0.0002 µg/L** (detection limit by ICP-MS/MS using a collision/reaction cell with 10% $CH_3F(g)$ and 90% $He(g)$ [306]) | **0.050 Excess** $\frac{\text{Cancer Deaths}}{1,000,000 \text{ People}}$ | 1 Excess $\frac{\text{Cancer Death}}{20,000,000 \text{ People}}$ |

dissociation, CID) or by slowing the polyatomic ion down (kinetic energy discrimination, KED) [300]. Or if a polyatomic ion exits an ICP, enters a C/RC, and collides with a reactive gas, such as hydrogen ($H_2(g)$), the interference can be eliminated by changing the mass of the polyatomic ion [300]. For example, an Agilent Technologies 7500c ICP-MS using a C/RC with 0.5 mL/minute of He(g) and 3.8 mL/minute of $H_2(g)$ eliminated interferences from 1 g/L of sodium chloride (NaCl) and gave a 0.025 µg/L detection limit for total As (Table 4) [302]. In this case, this detection limit is the lowest concentration that can be quantified at the 99.86% confidence level [302, 303]. This 0.025 µg/L detection limit is 400 times less than a 10 µg/L drinking water standard (Table 4).

Another advance uses flow injection analysis (FIA) and a knotted reactor to remove interference precursors from the sample matrix and concentrate total inorganic As before analysis by ICP-MS [304]. Arsenate (As(V)) was reduced to arsenite (As(III)) in a solution of 1% (mass/volume) L-cysteine ($HSCH_2CHNH_2COOH$) and 0.03 molar (M) nitric acid ($HNO_3$) [304]. This As(III) was complexed with a solution of 0.1% (mass/volume) ammonium pyrrolidine dithiocarbamate (($CH_2)_4NCS_2NH_4$) [304]. This complex was absorbed on the inner wall of a knotted reactor, in this case a 150-centimeter (cm) long by 0.5-millimeter (mm) inside diameter (ID) piece of polytetrafluoroethylene (PTFE) tubing [304]. The interference precursors from the sample matrix were washed away and the total inorganic As was concentrated when this complex was absorbed in the reactor [304]. This complex was desorbed from the reactor with 1 molar (M) $HNO_3$ and eluted into a Perkin-Elmer-Sciex ELAN 5000 ICP-MS

[304]. This gave a 0.029 μg/L detection limit for total inorganic As (Table 4) [304]. This detection limit is defined as 3 times the sample standard deviation ($s$), presumably from the measurement of reagent blanks [304].

**3.6.3 Inductively coupled plasma-tandem mass spectrometry.** Advances in inductively coupled plasma-tandem mass spectrometry (ICP-MS/MS) give detection limits for total As that are 0.01 μg/L or less [301]. The first MS is typically used as a mass filter and lets through only ions at $m/z$ = 75 ($^{75}As^+$(g), $^{40}Ar^{35}Cl^+$(g), $^{38}Ar^{37}Cl^+$(g), $^{37}Cl_2{}^1H^+$(g), $^{40}Ca^{35}Cl^+$(g), and $^{40}Ar^{34}S^1H^+$(g)) [299–301]. If present, these ions enter a C/RC and the $^{75}As^+$(g) selectively reacts with either oxygen ($O_2$(g)) or fluoromethane ($CH_3F$(g)) to produce $^{75}As^{16}O^+$(g) at $m/z$ = 91 or $^{75}As^{12}C^1H_2{}^+$(g) at $m/z$ = 89, respectively [301, 305, 306]. The second MS is used to filter the interferences at $m/z$ = 75, and quantify the $^{75}As^{16}O^+$(g) at $m/z$ = 91 or $^{75}As^{12}C^1H_2{}^+$(g) at $m/z$ = 89 [301, 305, 306]. This use of 2 mass spectrometers in tandem increases sensitivity and lowers the detection limit [301].

For example, if $O_2$(g) is used in the C/RC, a 0.0016 μg/L detection limit for total As was observed (Table 4) [306]. This detection limit is defined as the lowest concentration that can be quantified at the 99.7% confidence level [303, 305]. If 10% $CH_3F$(g) and 90% He(g) are used in the C/RC, a 0.0002 μg/L detection limit for total As was observed (Table 4) [306]. This detection limit is defined as $3s/m$, where $s$ is the sample standard deviation from 10 measurements of a reagent blank, and $m$ is the average slope from 10 calibration graphs [306].

ICP-MS/MS is a very new technology; the first commercial instruments were sold in 2012 [301]. As a result, ICP-MS/MS is mostly used for research and is not commonly used in drinking water testing laboratories. However, ICP-MS/MS will likely become more commonly used for drinking water testing in the future.

**3.6.4 Hydride generation-gas chromatography-photoionization detection.** Advances in hydride generation-gas chromatography-photoionization detection (HG-GC-PID) give a detection limit for total inorganic As at 0.00082 μg/L [307]. This detection limit is defined as $3s$, where $s$ is the sample standard deviation from 5 measurements of a reagent blank [307]. The hydride generation step uses 50 mL of sample, 2.0 mL of concentrated hydrochloric acid (HCl), 3.0 mL of 1.0 M potassium iodide (KI), and 4.0 mL of 4.0% (weight/volume) sodium borohydride ($NaBH_4$) to reduce arsenate (As(V)) and arsenite (As(III)) to arsine gas ($AsH_3$(g)) [307]. Helium (He(g)) is used as a carrier gas to move the $AsH_3$(g) from the reaction vessel, to a trap at −50°C, a trap at −196°C, a gas chromatograph (GC), and a photoionization detector (PID) [307]. If present, water vapor ($H_2O$(g)) is an interference and is removed from the $AsH_3$(g) in a trap that is cooled to −50°C with dry ice ($CO_2$(s)) and 2-propanol ($CH_3CHOHCH_3$) [307]. After this step, the $AsH_3$(g) is concentrated in a trap that is cooled to −196°C with liquid nitrogen ($N_2$(l)) [307]. If present, stibine gas ($SbH_3$(g)) is an interference and is separated from the $AsH_3$(g) in a GC with a Carbopack™ B HT column [307]. Finally, $AsH_3$(g) is quantified using a 10.2 electron volt (eV) PID [307].

**3.6.5 Summary of other advances in analytical chemistry methods.** In summary, no country needs to use the less protective 10 μg/L standard or guideline due to the expense of analytical chemistry methods. There are many methods for measuring total As to lower and the more protective concentrations (Table 4).

**3.6.6 Other advances in drinking water treatment technologies.** Another reason for continuing to use a 10 μg/L drinking water standard is "treatment performance" [18, 272]. However, recent advances in treatment technologies allow removal of As to concentrations that are significantly lower than 10 μg/L and would allow lower standards that would be more protective of public health. Selected examples of these advances follow.

By law, the Netherlands has a 10 μg/L drinking water standard [248]; however, the Netherlands voluntarily uses a less than 1 μg/L drinking water guideline to better protect public health. In 2015, "the Association of Dutch Drinking water Companies (Vewin) voluntarily

agreed on a guideline of <1 μg/L for As in drinking water" [308]. "This policy is based on a two-step assessment of As in drinking water, including i) an assessment of excess lung cancer risk for Dutch population and ii) a cost-comparison between the health care provision for lung cancer and As removal from water to avoid lung cancer" [308]. The average concentrations of As in raw water from 241 public supply well fields in the Netherlands ranges from <0.5 μg/L to 69 μg/L; the treatment of this water to <1 μg/L saves the Netherlands from 7.2 million Euros (M€)/year to 14 M€/year [310]. This 7.2 M€/year to 14 M€/year includes the savings in health care costs from not having to treat excess lung cancer cases and the engineering costs from treating drinking water [308].

In the Netherlands, As is typically removed from raw well water by an optimized aeration and rapid sand filtration process [308]. This aeration oxidizes the soluble Fe(II) and As(III) that is naturally found in raw well water to insoluble Fe(III) and As(V) [308]. A soluble Fe(III) coagulant, such as $FeCl_3$, is sometimes added to the raw well water [308]. This produces As-Fe (III) precipitates that are removed in a rapid sand filter [308]. This sand filter sometimes uses a coarse granular top layer and finer bottom layer [308]. This optimized process routinely removes As to <1 μg/L [308].

An optimized reverse osmosis process can also economically remove As from drinking water to concentrations that are significantly lower than 10 μg/L [309]. This process uses a 2-stage membrane cascade and can supply drinking water at 0.5 μg of As/L to a population of 20,000 people for US $1,041/day or US $0.52/m$^3$ [309]. A 2-stage membrane cascade has 2 reverse osmosis units connected in series. The feed water enters the first stage, the retentate or reject water from the first stage is discarded, and the permeate from the first stage enters the second stage. The retentate from the second stage is added to the feed water of the first stage. The permeate from the second stage is disinfected and distributed as drinking water. This process uses polyamide membranes [309]. The most significant cost is energy consumption; it is 35% of the total cost [309].

In summary, no country needs to use the less protective 10 μg/L standard or guideline due to limitations of treatment technologies. There are many treatment methods for reducing total As concentrations to lower more protective concentrations (Table 5).

**3.7 Implementation of regulations.** Unfortunately, having legal standards for drinking water quality does not guarantee that these standards will be implemented, or that efforts will be made by suppliers to meet those standards, especially in regions where government and resources are limited. Implementation of standards can also be complicated when multiple jurisdictions are involved [310]. Transparency is vital for the effectiveness of guidelines and standards to protect public health [310]. In countries where standards bureaus control drinking water standards and release them only for a fee, transparency is severely curtailed. Compliance requires additional efforts and resources.

In contrast, in regions where government is effective and resources are sufficient, such as in Denmark and the Netherlands, reducing the national drinking water standard for As to below 5 μg/L in was found to be both technically feasible and affordable [248, 311]. Even in regions in which there are limited resources for implementation and compliance, establishing and updating drinking water standards and guidelines can be of use to stakeholders who need to determine whether a water source is safe, and to put pressure on water suppliers to improve the quality of the water [311].

## 4. Conclusions

As demonstrated in this study, national and international guidelines and standards for As in drinking water need to be updated in accordance with current research and technologies.

**Table 5. Common drinking water standards, guidelines, and public health goals for total arsenic (As) in micrograms per liter (μg/L), the effluent concentrations of total As in μg/L from water treatment systems used in the high-income world, and the estimated cancer risks at these concentrations (Eqs 1, 2, 3, 4, 5 and 6).** These cancer risks are in bold font and rounded to 2 figures.

| Drinking Water Standard, Drinking Water Guideline, Public Health Goal, or Effluent Concentration | Number of Excess Cancer Deaths 1,000,000 People | 1 Excess Cancer Death Number of People |
|---|---|---|
| **50 μg/L** (drinking water standard common in lower income countries) | **12,000 Excess Cancer Deaths** 1,000,000 People | 1 Excess Cancer Death 81 People |
| **10 μg/L** (WHO provisional drinking water guideline; drinking water standard common in higher income countries) | **2,500 Excess Cancer Deaths** 1,000,000 People | 1 Excess Cancer Death 400 People |
| **<1 μg/L** (treatment performance using an optimized aeration and rapid sand filtration process [308]) | **< 250 Excess Cancer Deaths** 1,000,000 People | < 1 Excess Cancer Death 4,000 People |
| **0.5 μg/L** (treatment performance using an optimized reverse osmosis 2-stage membrane cascade [309]) | **120 Excess Cancer Deaths** 1,000,000 People | 1 Excess Cancer Death 8,100 People |
| **0.004 μg/L** (public health goal set by the California Environmental Protection Agency [8]) | **1.0 Excess Cancer Death** 1,000,000 People | 1 Excess Cancer Death 1,000,000 People |

Updating regulations has the potential to improve public health by reducing As exposures worldwide. Technologies are now available that could enable both resource-limited and high income countries to adopt more protective drinking water regulations for As.

The WHO 10 μg/L drinking water guideline for As is provisional because it does not sufficiently protect public health. The value of 10 μg/L was specified on the basis of "analytical achievability" or "treatment performance" [17, 18, 132, 272]. However, since the value for this guideline of 10 μg/L for was selected by the WHO in 1993, numerous new technologies and methods have been developed so that 10 μg/L no longer represents a practical limit for either analytical achievability or As treatment performance. This global drinking water guideline should be updated to better protect public health.

Many countries, especially lower income countries continue to use the long-outdated (1963) former WHO standard of 50 μg/L as a maximum allowable concentration for As [16]. Because many of these countries also have high populations, nearly one third of the world's population lives in jurisdictions with a 50 μg/L standard for As. This concentration of 50 μg/L is associated with high levels of morbidity and mortality and can no longer be justified by the high cost of As quantification or treatment since new low-cost analytical and treatment methods are now available. Lowering the maximum allowable concentration from 50 μg/L to 10 μg/L or lower is urgently needed to avoid countless preventable cancer deaths and to better protect public health.

A variety of solutions are required to update and set more protective national drinking water standards and guidelines for As. In fact, each country might set a hierarchy of drinking water standards and guidelines for As. For example, low-income countries with limited resources might need to rely on relatively inexpensive spectrophotometers for testing and relatively simple systems for treating drinking water. If such a country or parts of a country had As-free water (Figs 2 and 3), then a 50 μg/L standard based on analytical achievability could be lowered to a more protective 0.3 μg/L standard through simple testing and water sharing (Table 2). This would lower the lifetime cancer risk from 12,000 excess cancer deaths in 1,000,000 people (1 excess cancer death in 81 people) to 74 excess cancer deaths in 1,000,000 people (1 excess cancer death in 13,000 people; see Table 2). If such a country or parts of the

country did not have As-free water (Figs 2 and 3), then a 50 μg/L standard based on "treatment performance" could be lowered to a more protective <5 μg/L standard through treatment (Table 3). This would lower the lifetime cancer risk from 12,000 excess cancer deaths in 1,000,000 people (1 excess cancer death in 81 people) to <1,200 excess cancer deaths in 1,000,000 people (<1 excess cancer death in 810 people; see Table 3).

Countries with greater economic resources have access to more expensive instruments for testing and relatively sophisticated systems for treating drinking water. If such a country or parts of the country had As-free water (Figs 2 and 3), then a 10 μg/L standard based on analytical achievability could be lowered to a more protective ≤0.004 μg/L standard through testing and selective water use (Table 4). This would lower the lifetime cancer risk from 2,500 excess cancer deaths in 1,000,000 people (1 excess cancer death in 400 people) to ≤1 excess cancer deaths in 1,000,000 people (Table 4). If such country or parts of the country did not have As-free water (Figs 2 and 3), then a 10 μg/L standard based on treatment performance to a more protective 0.5 μg/L standard through more complete As treatment (Table 5). This would lower the lifetime cancer risk from 2,500 excess cancer deaths in 1,000,000 people (1 excess cancer death in 400 people) to 120 excess cancer deaths in 1,000,000 people (1 excess cancer death in 8,100 people; see Table 5).

## Supporting information

**S1 File. Arsenic regulations, populations, GDPs, and reference links for 195 countries.** (XLSX)

## Acknowledgments

We are grateful to Leif Rasmussen, Esq. for his assistance with international law. We are also grateful to Susan Murcott and Prakriti Sardana for their assistance reviewing this manuscript and to Mohammad Yusuf Siddiq for help translating Arabic documents.

## Author Contributions

**Conceptualization:** Seth H. Frisbie.

**Data curation:** Seth H. Frisbie, Erika J. Mitchell.

**Formal analysis:** Seth H. Frisbie, Erika J. Mitchell.

**Funding acquisition:** Seth H. Frisbie.

**Investigation:** Seth H. Frisbie, Erika J. Mitchell.

**Methodology:** Seth H. Frisbie, Erika J. Mitchell.

**Project administration:** Seth H. Frisbie.

**Validation:** Seth H. Frisbie, Erika J. Mitchell.

**Visualization:** Seth H. Frisbie, Erika J. Mitchell.

**Writing – original draft:** Seth H. Frisbie, Erika J. Mitchell.

**Writing – review & editing:** Seth H. Frisbie, Erika J. Mitchell.

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
