## [Decision Letter · Decision Letter 0]

1 Dec 2021

PONE-D-21-35059Arsenic in drinking water: an analysis of global drinking water regulations and recommendations for updates to protect public healthPLOS ONE

Dear Dr. Frisbie,

Thank you for submitting your manuscript to PLOS ONE. After careful consideration, we feel that it has merit but does not fully meet PLOS ONE’s publication criteria as it currently stands. Therefore, we invite you to submit a revised version of the manuscript that addresses the points raised during the review process.

The manuscript presents a thoughtful review of global policies, potential exposures, and future avenues of advancement. We ask that you would please address some comments brought up in review to further strengthen the manuscript.

We look forward to receiving your revised manuscript.

Kind regards,

Aaron Specht

Academic Editor

PLOS ONE

Journal Requirements:

1. Thank you for stating the following in the Competing Interests/Financial Disclosure:

(TPH has provided expert witness testimony on the subject of prevention of falls in hospitals for K&L Gates Law Firm and Minter Ellison Law Firm.)

We note that you received funding from a commercial source: (K&L Gates Law Firm, Minter Ellison Law Firm)

Please respond by return email with your amended Competing Interests Statement and we will change the online submission form on your behalf.

3.  We note that Figures 4 and 5 in your submission contain map images which may be copyrighted. All PLOS content is published under the Creative Commons Attribution License (CC BY 4.0), which means that the manuscript, images, and Supporting Information files will be freely available online, and any third party is permitted to access, download, copy, distribute, and use these materials in any way, even commercially, with proper attribution. For these reasons, we cannot publish previously copyrighted maps or satellite images created using proprietary data, such as Google software (Google Maps, Street View, and Earth). For more information, see our copyright guidelines: http://journals.plos.org/plosone/s/licenses-and-copyright.

A. You may seek permission from the original copyright holder of Figure 4 and 5 to publish the content specifically under the CC BY 4.0 license. 

B. If you are unable to obtain permission from the original copyright holder to publish these figures under the CC BY 4.0 license or if the copyright holder’s requirements are incompatible with the CC BY 4.0 license, please either i) remove the figure or ii) supply a replacement figure that complies with the CC BY 4.0 license. Please check copyright information on all replacement figures and update the figure caption with source information. If applicable, please specify in the figure caption text when a figure is similar but not identical to the original image and is therefore for illustrative purposes only.

Additional Editor Comments:

The manuscript submitted has merit in the field and serves as an excellent reference of current situations globally with respect to Arsenic exposures and the story leading us to where we are today. I do think there should be some substantive revisions of sections brought up by reviews, but the overall content of the manuscript remains sounds. Thank you for your submission!

Reviewers' comments:

Reviewer's Responses to Questions

**Comments to the Author**

1. Is the manuscript technically sound, and do the data support the conclusions?

Reviewer #1: Yes

Reviewer #2: No

2. Has the statistical analysis been performed appropriately and rigorously? 

Reviewer #1: Yes

Reviewer #2: No

3. Have the authors made all data underlying the findings in their manuscript fully available?

Reviewer #1: Yes

Reviewer #2: Yes

4. Is the manuscript presented in an intelligible fashion and written in standard English?

Reviewer #1: Yes

Reviewer #2: No

5. Review Comments to the Author

Reviewer #1: the title of a project are compatible with their work results, and the idea of the project is very crucial, in particular, the effects of Arsenic element on public health, and the WHO always promote the researchers to focus that kind of study

Reviewer #2: Arsenic in drinking water: an analysis of global drinking water regulations and recommendations for updates to protect public health

Comments for the authors

In this study, the authors collated and analyzed all drinking water regulations for As from national governments worldwide and related them with the health risks. The data was not interpreted scientifically. The presentation of the data is poor.

Abstract: The authors have only mentioned the data collection and analysis. what is the purpose of the study? what are the recommendations from this study.? Please revise the abstract.

Please revise the very first statement of the introduction ‘…that is often found in groundwater wells’

226,000,000 people… 226 million people

many low-and middle-income countries still use the World Health Organization’s (WHO’s) 1963 drinking water standard... I don’t agree with this assumption. From the previous few decades, groundwater As has gained considerable attention and most of the literature data present the comparison with WHO 10 µg/L.

I think the authors should pay more attention to the regulation limits. The Department of Environmental Protection for New Jersey has proposed the As limit of 5 µg/L in drinking water.

There are several other threshold limits from different areas that are not mentioned in this review.

Moreover, these regulation limits are made depending on the hydrogeochemistry of the specific region. Depending on the exposure scenario, the health risk indices and threshold guidelines are formed. The authors completely missed this type of useful information.

Section 2: It seems like the authors have collected data only from US states. Please clarify.

Section 2.1: I suggest avoiding the use of authorial (we or I). The use of passive voice can be suitable.

This study is more like a meta-analysis review. The authors have collected abundant data and categorized it according to the threshold value, income, health, etc. I question the robustness of statistics.

Which software was used to extract values of the data plotted in a graph? The authors did not follow guidelines of the analysis used in societies in academic fields.

The authors actually used very simple calculations based on simple empirical equations (just simple calculations with Excel). Though they provided those values, I do not think it has much meaning and much scientific advancement.

I think they just collected data but did not interpret it in a mechanistic way.

Section 3.4: All of the data processing and calculation should be mentioned under the methodology section.

Section 3.5: the section seems totally irrelevant to the current study. how can you relate this section with the data collected from various studies? Did the authors categorize the analysis techniques also…??? Even the developing countries have advanced methods of As analysis.

Section 3.5.2: We can not start any new section with ‘for example’…example of what?

Table 2: how the data was verified. The consumption of As-contaminated water could not be the only reason for developing cancer.

It is ambiguous to relate the cancer data with instrument detection limits.

The conclusion is simply a summary of the results. What is the take-home message of this study?

All the figure captions and indications are wrongly mentioned in the manuscript.

The figures related to the structure of the dyes or molecules seem irrelevant to the study.

There are several grammatical and punctuation mistakes in the manuscript.

6. PLOS authors have the option to publish the peer review history of their article (what does this mean?). If published, this will include your full peer review and any attached files.

Reviewer #1: No

Reviewer #2: No

---

## [Author Response · Author response to Decision Letter 0]

6 Dec 2021

Journal Requirements: 

1. Thank you for stating the following in the Competing Interests/Financial Disclosure:

(TPH has provided expert witness testimony on the subject of prevention of falls in hospitals for K&L Gates Law Firm and Minter Ellison Law Firm.)

We note that you received funding from a commercial source: (K&L Gates Law Firm, Minter Ellison Law Firm)

Please respond by return email with your amended Competing Interests Statement and we will change the online submission form on your behalf.

-------------------

Response

The references to K&L Gates Law Firm, Minter Ellison Law Firm are likely for another manuscript. We have never heard of these firms and we do not have an author with the initials TPH. We do not have any commercial funders so we believe that we do not need to amend our Competing Interests Statement.

No changes made.

----------------

Response

All data are included in the supplementary file “supplementary file s1.xlsx”. This file is referenced with its caption on the very last line of our document, immediately following the References section.

No changes made.

-------------------

3. We note that Figures 4 and 5 in your submission contain map images which may be copyrighted. All PLOS content is published under the Creative Commons Attribution License (CC BY 4.0), which means that the manuscript, images, and Supporting Information files will be freely available online, and any third party is permitted to access, download, copy, distribute, and use these materials in any way, even commercially, with proper attribution. For these reasons, we cannot publish previously copyrighted maps or satellite images created using proprietary data, such as Google software (Google Maps, Street View, and Earth). For more information, see our copyright guidelines: http://journals.plos.org/plosone/s/licenses-and-copyright.

A. You may seek permission from the original copyright holder of Figure 4 and 5 to publish the content specifically under the CC BY 4.0 license. 

B. If you are unable to obtain permission from the original copyright holder to publish these figures under the CC BY 4.0 license or if the copyright holder’s requirements are incompatible with the CC BY 4.0 license, please either i) remove the figure or ii) supply a replacement figure that complies with the CC BY 4.0 license. Please check copyright information on all replacement figures and update the figure caption with source information. If applicable, please specify in the figure caption text when a figure is similar but not identical to the original image and is therefore for illustrative purposes only.

Response:

We created the maps in Figures 4 and 5 ourselves using the ggmap package in R. As requested in the reference document for the ggmap package (https://cran.r-project.org/web/packages/ggmap/citation.html), we have cited the authors of the ggmap package (Kahle and Wickham, 2013) since they provided the source code for the base maps on a CC by 4.0 basis. The ggmap source code was based on the public domain Natural Earth project. The figure captions of our maps cite Kahle and Wickham, 2013 as reference [264] as the source for these maps. These maps were not made with any copyrighted materials and they are suitable for CC by 4.0 licensing as is. We revised Section 2.2 (Data analysis and statistics) to explicitly state that we created the maps in R with the ggmaps package.

Response:

The caption for our Supporting Information file appears on the very last line of our document, immediately following the References section.

No changes made.

-------------------

-------------------

Response:

We have reviewed the reference list and it is complete and correct. We have not cited any retracted papers. We have fixed several typos and added missing doi links where needed. We deleted two references in response to reviewer comments and renumbered subsequent references.

Additional Editor Comments:

The manuscript submitted has merit in the field and serves as an excellent reference of current situations globally with respect to Arsenic exposures and the story leading us to where we are today. I do think there should be some substantive revisions of sections brought up by reviews, but the overall content of the manuscript remains sounds. Thank you for your submission!

Response:

Thank you for reading the manuscript carefully and soliciting helpful reviewer comments. This was a monumental effort, and we are grateful that our research will be made available with the publication of this paper for others to use.

Reviewers' comments:

Reviewer's Responses to Questions

Comments to the Author

1. Is the manuscript technically sound, and do the data support the conclusions?

Reviewer #1: Yes

Reviewer #2: No

2. Has the statistical analysis been performed appropriately and rigorously? 

Reviewer #1: Yes

Reviewer #2: No

3. Have the authors made all data underlying the findings in their manuscript fully available?

Reviewer #1: Yes

Reviewer #2: Yes

4. Is the manuscript presented in an intelligible fashion and written in standard English?

Reviewer #1: Yes

Reviewer #2: No

5. Review Comments to the Author

Reviewer #1: the title of a project are compatible with their work results, and the idea of the project is very crucial, in particular, the effects of Arsenic element on public health, and the WHO always promote the researchers to focus that kind of study

Reviewer #2: Arsenic in drinking water: an analysis of global drinking water regulations and recommendations for updates to protect public health

Comments for the authors

In this study, the authors collated and analyzed all drinking water regulations for As from national governments worldwide and related them with the health risks. The data was not interpreted scientifically. The presentation of the data is poor.

Abstract: The authors have only mentioned the data collection and analysis. what is the purpose of the study? what are the recommendations from this study.? Please revise the abstract.

Response:

We have revised the abstract to state the purpose and recommendations of the study more explicitly.

-----------------

Please revise the very first statement of the introduction ‘…that is often found in groundwater wells’

-------------------

Response:

Done.

-------------------

226,000,000 people… 226 million people

Response:

Done.

-------------------

many low-and middle-income countries still use the World Health Organization’s (WHO’s) 1963 drinking water standard... I don’t agree with this assumption. 

Response:

Please note that this is not an assumption. It is a fact that is directly supported by the data that we collated for this study. See our data in Table 1 that show that as of 2021, 36 countries still use the WHO’s 1963 drinking water standard. All but 2 of these countries are classified as low or middle income by the World Bank. As mentioned in the Abstract, the percentage of population living in countries that do not meet the current WHO drinking water guideline for As constitutes 32% of the total global population.

No changes made.

------------------

From the previous few decades, groundwater As has gained considerable attention and most of the literature data present the comparison with WHO 10 µg/L.

Response:

It is fortunate that most studies in the literature use the current WHO 10 µg/L guideline for comparison. However, many countries still legally use the former 1963 WHO guideline, as shown in Table 1 and explained in our previous response.

No changes made.

I think the authors should pay more attention to the regulation limits. The Department of Environmental Protection for New Jersey has proposed the As limit of 5 µg/L in drinking water.

There are several other threshold limits from different areas that are not mentioned in this review.

Response:

The title of this paper is “Arsenic in drinking water: an analysis of global drinking water regulations and recommendations for updates to protect public health.”. This paper focuses on national regulations, not the regulations of individual states, provinces, districts, etc. within any one nation such as the United States. Thus, we do not cover regulations for the individual states of the United States. We do cover As regulations for every national government in the world that we could locate in our extensive search for national regulations.

No changes made.

-------------------

Moreover, these regulation limits are made depending on the hydrogeochemistry of the specific region. Depending on the exposure scenario, the health risk indices and threshold guidelines are formed. The authors completely missed this type of useful information.

-------------------

Response:

As discussed in our Abstract, Introduction, and Section 3.3 (The drinking water guideline set by the World Health Organization), the World Health Organization’s (WHO’s) drinking water guidelines take into account health risk assessments, analytical capabilities, and treatment performance. The WHO guidelines are not based on hydrogeochemistry. More specifically, the 1963 WHO drinking water standard for As of 50 micrograms per liter (µg/L) was based entirely on “analytical achievability”. Similarly, the WHO’s 10 µg/L guideline established in 1993 was also based entirely on “analytical achievability”; however, in 2011 the additional proviso of “treatment performance” was added to the current WHO WHO’s 10 µg/L guideline. The WHO also states that these values cause an unacceptable risk of death from “skin and possibly other cancers” and cannot be set at a lower, more protective health-based level because it cannot be measured in a routine testing laboratory or removed by conventional treatment plants.

Furthermore, in Section 3.3, we wrote “National regulators are encouraged [by the WHO] to take local conditions, resources, and hazards into account when setting national standards [268,269].” We cited the WHO’s 2018 document “Developing drinking-water quality regulations” [our reference 269] in support of this statement.

No changes made.

------------------

Section 2: It seems like the authors have collected data only from US states. Please clarify.

Response:

The title of this paper is “Arsenic in drinking water: an analysis of global drinking water regulations and recommendations for updates to protect public health”. There is no suggestion in the title that any regulations from individual US states will be discussed, nor is there any survey of data from US states.

The title of our Section 2.1 is “International drinking water standards for arsenic”. As detailed in the first 2 sentences of that section, the entries in the database that we created and present in Table 1 include the 193 United Nations member states plus Taiwan and Kosovo. Individual US state regulations were not part of this database.

No changes made.

-------------------

Section 2.1: I suggest avoiding the use of authorial (we or I). The use of passive voice can be suitable.

Response:

Although PLOS does not have explicit advice regarding passive or active voice, many other leading publishers such as Sage Journals (https://journals.sagepub.com/author-instructions/phr), Cambridge University Press (https://www.cambridge.org/core/journals/enterprise-and-society/information/instructions-contributors), NatureProtocols (https://www.nature.com/nprot/for-authors/preparing-your-submission), and the Canadian Medical Association Journal (https://www.cmaj.ca/submission-guidelines) all explicitly recommend or require active voice rather than passive voice. We have not encountered any scientific journals that require passive voice to avoid agency. Thus, we believe it is acceptable to use active voice in scientific writing.

No changes made.

This study is more like a meta-analysis review. The authors have collected abundant data and categorized it according to the threshold value, income, health, etc. I question the robustness of statistics.

Which software was used to extract values of the data plotted in a graph?

Response:

As stated in Section 2.2 (Data analysis and statistics), we used R version 4.1.1 for all statistics and data analysis. All of the figures and maps were also made with R. We added a sentence to the end of Section 2.2 to explicitly state that we used R to create the figures and maps as well as to perform the statistical analyses.

 The authors did not follow guidelines of the analysis used in societies in academic fields.

The authors actually used very simple calculations based on simple empirical equations (just simple calculations with Excel). Though they provided those values, I do not think it has much meaning and much scientific advancement.

I think they just collected data but did not interpret it in a mechanistic way.

-------------------

Response:

As noted in Section 2.2 (Data analysis and statistics), we used R for the statistical calculations, not Excel. We believe our use of statistics is in accordance with standard practices for scientific writing. After comparing our reporting of statistics to the PLOS guidelines (https://journals.plos.org/plosone/s/submission-guidelines.#loc-statistical-reporting), to be fully in compliance, we added a statement to Section 2.2 about why we did not use corrections for multiple comparisons. That is, we did not use corrections for multiple comparisons in this paper because we did not make multiple statistical comparisons of the data.

As noted in Sections 3.1 and 3.2, this study is the most comprehensive study of national drinking water regulations for arsenic that has ever been published to date. It provides and analyses data for almost twice as many countries (195) as a 2018 WHO survey (104 countries). As a result, our Table 1 and the references cited in the table constitute the most complete database of national drinking water regulations published to date. By virtue of its completeness of scope, this study provides comparative information about the drinking water regulations for arsenic worldwide that has never been available before.

The statistical analyses of this data set provide the first firm data concerning the relationship between the protectiveness of national regulations for arsenic in drinking water and national per capita income. More specifically, “the GDPs per capita of countries with As regulations equal to or more protective than the WHO of guideline of 10 µg/L were significantly higher (n = 129, M = $17,678) than those of countries with As regulations less protective than the current 10 µg/L WHO guideline (n = 36, M = $5,384) (F(2,177) = 7.55, p<.001)”. It also shows that the recency of regulations is not related to national per capita income. An extremely important finding of this study is that nearly 1/3 of the world’s population live in jurisdictions that still use the 1963 WHO standard of 50 μg As/L as their national standard. We believe these findings are crucial for understanding the relationship between regulations and risks to public health, and that they are critical for directing future efforts to protect health.

No changes made.

-------------------

Section 3.4: All of the data processing and calculation should be mentioned under the methodology section.

----------------

Response:

This Section, (3.4 Health-based drinking water guidelines) of the Results and Discussion section, provides discussion of several regulatory documents and their implications for public health. The equations in this section were not part of our research on current international drinking water regulations, so it is not appropriate to discuss the calculations in the Materials and Methods section, which describes the materials and methods for our collation of international drinking water regulations and their analysis.

The equations are provided here for completeness and transparency to demonstrate the implications of regulations that are based on “analytical achievability”, “treatment performance”, the risk of death from various cancers, and public health. This is essential for the following discussion on various tangible ways to reduce the risk of death from cancer and protect public health.

No changes made.

-----------------

Section 3.5: the section seems totally irrelevant to the current study. how can you relate this section with the data collected from various studies? Did the authors categorize the analysis techniques also…??? Even the developing countries have advanced methods of As analysis.

------------------

Response:

As stressed in Section 4 (Conclusions), our overarching recommendation is that international guidelines and national standards for As in drinking water are out of date and need to be updated in accordance with the advances of science.

In Section 3.2 (Arsenic regulations, population, and per capita income), we showed that many countries, especially low and medium income countries, still use the 1963 WHO standard of 50 μg As/L as their national standards. This standard is based on analytical achievability from 58 years ago. In Section 3.5 (Low-cost methods for improving public health by reducing arsenic exposure), we enumerate examples of currently available low-cost methods for analysis and treatment of As contaminated water. This enumeration is an essential step for our argument that it is feasible for low and middle income countries to adopt more stringent standards regarding As in drinking water. Again, approximately one-third of the global population live in countries that do not meet the current 10 µg/L WHO drinking water guideline for As. We believe that reducing the income inequities in international drinking water regulations would be a positive step towards reducing the global health gap for lower income people.

No changes made.

-----------------

Section 3.5.2: We can not start any new section with ‘for example’…example of what?

----------------

Response:

We have reworded this sentence for clarity.

----------------

Table 2: how the data was verified. 

Response:

The comparison figures in the 2 right-hand columns of Table 2 are drawn from equations 1-6, as stated in the table caption. As described in detail in Section 3.4 (Health-based drinking water guidelines), these equations and estimates of cancer deaths are drawn directly from published regulatory documents, for which we provide citations in Section 3.4.

No changes made.

The consumption of As-contaminated water could not be the only reason for developing cancer.

-------------------

Response: 

It is quite correct that “The consumption of As-contaminated water could not be the only reason for developing cancer”. This is why regulators rely on the concept of “Excess cancer deaths”, a common metric in risk assessment used for calculating health-based drinking water quality guidelines and standards. We revised our description of excess cancer deaths in Section 3.4 (Health-based drinking water guidelines) to clarify this point.

-----------------

It is ambiguous to relate the cancer data with instrument detection limits.

Response:

The purpose of Table 2 is to facilitate direct comparisons between regulatory limits, detection limits, and regulatory risk assessments. We believe these comparisons will be useful for prioritizing efforts to protect public health.

No changes made.

----------------

The conclusion is simply a summary of the results. What is the take-home message of this study?

-------------------

Response:

We have added a new paragraph to Section 4 (Conclusions) to state our take-home message more explicitly.

All the figure captions and indications are wrongly mentioned in the manuscript.

--------------

Response:

We have checked the figure captions and in-text citations and verified that they are in fully in compliance with PLOS guidelines (https://journals.plos.org/plosone/s/figures). Two figures were numbered incorrectly; they have been deleted in accordance with the following comment.

-------------------

The figures related to the structure of the dyes or molecules seem irrelevant to the study.

Response:

We have deleted these figures and their associated citations.

There are several grammatical and punctuation mistakes in the manuscript.

------------------

Response:

Thank you for this observation. We have proofread the document again and fixed several typographical, grammar, and punctuation errors.

---

## [Decision Letter · Decision Letter 1]

21 Jan 2022

Arsenic in drinking water: an analysis of global drinking water regulations and recommendations for updates to protect public health

PONE-D-21-35059R1

Dear Dr. Frisbie,

We’re pleased to inform you that your manuscript has been judged scientifically suitable for publication and will be formally accepted for publication once it meets all outstanding technical requirements.

Kind regards,

Aaron Specht

Academic Editor

PLOS ONE

Additional Editor Comments (optional):

Reviewers' comments:

Comments were addressed and manuscript is ready for publication

---

## [Editor Report · Acceptance letter]

11 Mar 2022

PONE-D-21-35059R1 

Arsenic in drinking water: an analysis of global drinking water regulations and recommendations for updates to protect public health 

Dear Dr. Frisbie:

I'm pleased to inform you that your manuscript has been deemed suitable for publication in PLOS ONE. Congratulations! Your manuscript is now with our production department. 

Kind regards, 

on behalf of

Dr. Aaron Specht 

Academic Editor

PLOS ONE